# Single-protein/RNA imaging reveals ZNF598 as a limiting factor in resolving collided ribosomes

Ana C De La Cruz[1,2], Garrett Tisdale[1,2], Emily Nakayama[1,2], Zhiyuan Huang[1,2], Niladri K Sinha[3,4], Rachel Green [3,4] & Bin Wu [1,2,5✉]

## Abstract

Ribosome-associated protein quality control (RQC) is a surveillance system that identifies and processes aberrant mRNAs with collided ribosomes. ZNF598 plays a key role by ubiquitinating the 40S subunit of collided ribosomes. However, how ZNF598 distinguishes stalled from transient ribosome collisions remains unclear. To address this, we developed a method to visualize the binding of a single protein to a specific mRNA while simultaneously determining its translation status. By endogenously tagging ZNF598 with HaloTag, we observed its strong interaction with RQC reporter mRNAs. We discovered that multiple ZNF598s engage with a single RQC mRNA, suggesting that ZNF598 recognizes more than just the leading collided ribosome in a queue. Overexpressing ZNF598 increased the ribosomal clearance rate, indicating that it is a rate-limiting factor for RQC. Interestingly, a subset of supposedly "normal" mRNAs may be damaged and targeted by ZNF598, underscoring the importance of RQC to maintain the proteome quality even in unstressed conditions. Under global UV-induced RNA damage, ZNF598 recruitment to the reporter RQC mRNA diminished, highlighting its role as a limiting factor in managing widespread ribosome collisions.

**Keywords** Protein–RNA Interactions; Ribosome-associated Quality Control; RNA; Single Molecule; Translation
**Subject Category** Translation & Protein Quality

## Introduction

Proteins play essential roles in the metabolism of RNAs (Hentze et al, 2018). Starting from transcription, various protein factors bind, process, unwind, and covalently modify RNAs to regulate their fates. Traditional biochemical approaches, while valuable for defining protein–RNA interactions, often require breaking up the cells and thus introduce some limitations (Ramanathan et al, 2019).

For example, interactions may be nonspecific or transient and may not survive cell lysis once removed from the cellular context. More importantly, such methods provide only a snapshot of the interaction, leaving the dynamics and functional consequences to be independently validated. Single-molecule imaging in live or fixed cells has emerged as a powerful tool to characterize protein–RNA interactions in their native environment, addressing some of these limitations. We have established dual-color fluorescence fluctuation spectroscopy to measure the interactions between protein and mRNA by analyzing the correlated fluorescence signals of the interacting complex (Wu et al, 2015a). However, the method has primarily been applied to relatively abundant mRNAs, like β-actin, as it requires averaging signals from many molecules. High-precision single-molecule fluorescence in situ hybridization and immunofluorescence (smFISH-IF) has been applied to analyze protein–RNA interactions in cells (Eliscovich et al, 2017). Similarly, protein–RNA interactions are visualized in an RNA fluorescence three-hybrid system in live cells by anchoring RNAs to a subcellular structure to enrich target fluorescent RBPs (Duan et al, 2021). However, these methods cannot monitor the functional outcome of protein binding. Therefore, it is crucial to directly visualize single-protein–RNA interactions in live cells on functional RNA substrates. This is a challenging task, akin to observing the effect of single transcription factors or chromatin modifiers on the transcription dynamics of specific genes (Morisaki et al, 2014; Donovan et al, 2019; Nguyen et al, 2021; Paakinaho et al, 2017; Wang et al, 2016b).

To visualize the translation dynamics of single mRNAs, we and others have developed Single-Molecule Imaging of Nascent PeptideS (SINAPS), which uses fluorescently labeled antibody fragments to tag newly synthesized arrays of peptide epitopes (Wu et al, 2016; Yan et al, 2016; Morisaki et al, 2016; Pichon et al, 2016; Wang et al, 2016a). Recently, we improved an existing membrane tethering system that localizes the SINAPS reporter mRNAs to the plasma membrane, allowing us to track the translation of single mRNAs for hours with total internal reflection microscopy (TIRFM) (Yan et al, 2016; Livingston et al, 2023). A much-anticipated advancement in the field is to observe single-protein factors on reporter mRNAs while simultaneously monitoring their translation dynamics. In this study, we fill this gap by using a similar membrane tethering system.

[1]Department of Biophysics and Biophysical Chemistry, Johns Hopkins University School of Medicine, Baltimore, MD 21205, USA. [2]The Center for Cell Dynamics, Johns Hopkins University School of Medicine, Baltimore, MD 21205, USA. [3]Department of Molecular Biology and Genetics, Johns Hopkins University School of Medicine, Baltimore, MD 21205, USA. [4]Howard Hughes Medical Institute, Chevy Chase, MD 20815, USA. [5]The Solomon H Snyder Department of Neuroscience, Johns Hopkins University School of Medicine, Baltimore, MD 21205, USA. ✉E-mail: bwu20@jhmi.edu

Genetic mutations, premature polyadenylation or exposure to damaging agents can produce defective mRNAs. Translation of these mRNAs may result in misfolded or malfunctioning proteins, potentially leading to debilitating neurodegenerative disorders, such as Alzheimer's, Parkinson's diseases or Amyotrophic Lateral Sclerosis (ALS) (Wu et al, 2019; Rimal et al, 2021; Latallo et al, 2023). To counteract this, cells have evolved quality control mechanisms that target these defective mRNAs and their aberrant protein products. One such system is the ribosome-associated protein quality control (RQC) pathway, triggered by collided ribosomes on defective mRNAs (Filbeck et al, 2022; Sitron and Brandman, 2020; Joazeiro, 2019; Inada, 2020). The pathway leads to ribosome disassembly, recycling, and the degradation of the damaged mRNA and truncated proteins (Brandman and Hegde, 2016). One well-studied example of a strong RQC substrate is an mRNA containing a long stretch of a poly(A) sequence in the open reading frame (ORF) to mimic a prematurely polyadenylated mRNA. In mammalian cells, the E3 ubiquitin ligase, ZNF598 (Hel2 in yeast) is essential for rescuing stalled ribosomes on poly(A) sequences by recognizing the distinct 40S–40S interface of collided ribosomes (Garzia et al, 2017; Sundaramoorthy et al, 2017; Juszkiewicz and Hegde, 2017; Juszkiewicz et al, 2018). ZNF598 initiates the RQC pathway by ubiquitinating small ribosomal proteins (Garzia et al, 2017; Sundaramoorthy et al, 2017; Juszkiewicz and Hegde, 2017; Juszkiewicz et al, 2018; Chandrase-karan et al, 2019; Ikeuchi et al, 2019; Matsuo et al, 2017). Following ubiquitination, ribosome disassembly is facilitated by the RQT (RQC-trigger) complex (Matsuo et al, 2017; Juszkiewicz et al, 2020; Hashimoto et al, 2020; Matsuo et al, 2020; Narita et al, 2022). ASCC3, an ATP-dependent helicase, is essential for splitting stalled ribosomes in a ZNF598-dependent manner (Juszkiewicz et al, 2020). Subsequently, the 60S ribosome subunit, still associated with the nascent peptide-tRNA complex, is targeted by downstream RQC factors to degrade the nascent protein (Shao et al, 2013; Bengtson and Joazeiro, 2010; Defenouillère et al, 2013; Lyumkis et al, 2014; Shao et al, 2015).

We previously constructed an RQC SINAPS reporter with a poly(A) stalling sequence at the end of the ORF (Goldman et al, 2021). Our findings highlighted that ribosome stalling leads to queues of collided ribosomes and that the clearance of stalled ribosomes is slow compared to translation elongation and termination. Depletion of the collision sensor ZNF598 led to a slowdown in ribosome clearance, resulting in an even longer queue. These data indicated that the primary bottleneck for the disassembly of collided ribosomes is recognition of the collision, with ZNF598 being an essential factor. However, it remained unclear whether ZNF598 activity is rate-limiting and whether it only targets the leading collided ribosomes.

In this study, we applied live-cell imaging to investigate the interaction between ZNF598 and mRNAs. Employing CRISPR/Cas9-mediated knock-in technology, we generated a homozygous ZNF598-HaloTag cell line and confirmed its functionality. We then incorporated the SINAPS system for live-cell imaging and investigated the interaction of the endogenously tagged ZNF598 with both poly(A) and "normal" messages. Our results demonstrated that ZNF598 interacts strongly with collided ribosomes on poly(A) mRNAs, with a stoichiometry greater than one. Over-expression of ZNF598 significantly increases the clearance speed from the ribosomal queues, establishing ZNF598 as a rate-limiting

factor in the RQC pathway. Surprisingly, we discovered that a fraction of "normal" messages contain collided ribosomes and recruit ZNF598. To explore this further, we applied UV radiation to globally damage mRNAs and measured the interaction of reporter mRNAs with ZNF598. Our data confirmed that ZNF598 is a limiting factor for mitigating the impact of widespread insults. Overall, we developed an assay to track single-protein–mRNA interactions and the translation status in real-time, which opens the door to quantitatively evaluate the effect of regulatory protein factors on mRNA metabolism. By applying the technology to RQC, we revealed the critical role of ZNF598 as a crucial limiting factor in the disassembly of collided ribosomes.

## Results

### Imaging single-protein factors interacting with mRNAs in live cells

Despite significant advancements in single-molecule visualization technologies, tracking single RBPs on specific, freely diffusing mRNAs in cells remains a technical challenge. Non-interacting RBPs produce a high background, overwhelming the signal from a single fluorophore. Common tricks for single-molecule detection in live cells often fail in this context. For example, using a long exposure time to accumulate signals from stationary molecules over a diffusive background is ineffective because mRNAs are also mobile. Similarly, sparse labeling or photoactivation is unhelpful because stochastically visible proteins are unlikely to be on the designated mRNAs.

To overcome these challenges, we implemented a previously developed system to tether mRNAs to the plasma membrane, allowing for extended imaging periods (Yan et al, 2016; Livingston et al, 2023; Goldman et al, 2021). For this, we constructed mRNA reporters with 24xMS2 binding sites (MBS) inserted into the 3′ untranslated region (UTR). An MS2-coat protein (MCP) fused to a CAAX motif was co-expressed in cells with the reporter, enabling the anchoring of the mRNA to the plasma membrane (Wu et al, 2012, 2015b). Specifically, we used a synonymized tandem MCP (tdMCP) fused with mScarlet Red Fluorescent Protein (Bindels et al, 2017) (tdMCP-mScarlet-CAAX) for RNA labeling. Tethering mRNAs to the plasma membrane offers two advantages for single-molecule detection. First, it allows TIRFM to selectively illuminate the mRNA and the associated RBP tethered on the membrane, while rejecting signal from the bulk cytoplasmic proteins (Fig. 1A). Second, it limits the diffusion of mRNAs, facilitating a long exposure time to integrate the bound RBP signal while averaging out the diffusive background signals. To further enhance single RBP detection, we employed the widely used "HaloTag", and covalently labeled it with the bright and photostable Janelia Fluorophores (Los et al, 2008; Grimm et al, 2017, 2016).

To model single RBPs binding to mRNAs, we utilized a well-characterized orthogonal PP7 stem-loop system. We inserted different numbers of PP7 Binding Sites (PBS) before the MBS in the 3′UTR of the reporter mRNA (Fig. 1B) and then used the tandem PP7 coat protein fused to HaloTag (tdPCP-HaloTag), which binds PBS with nanomolar affinity, as a model RBP (Wu et al, 2012, 2015b; Chao et al, 2008). We constructed reporter mRNAs: Fluc-nxPBS-24xMBS ($n = 0$, 1, 3 or 12), where Fluc

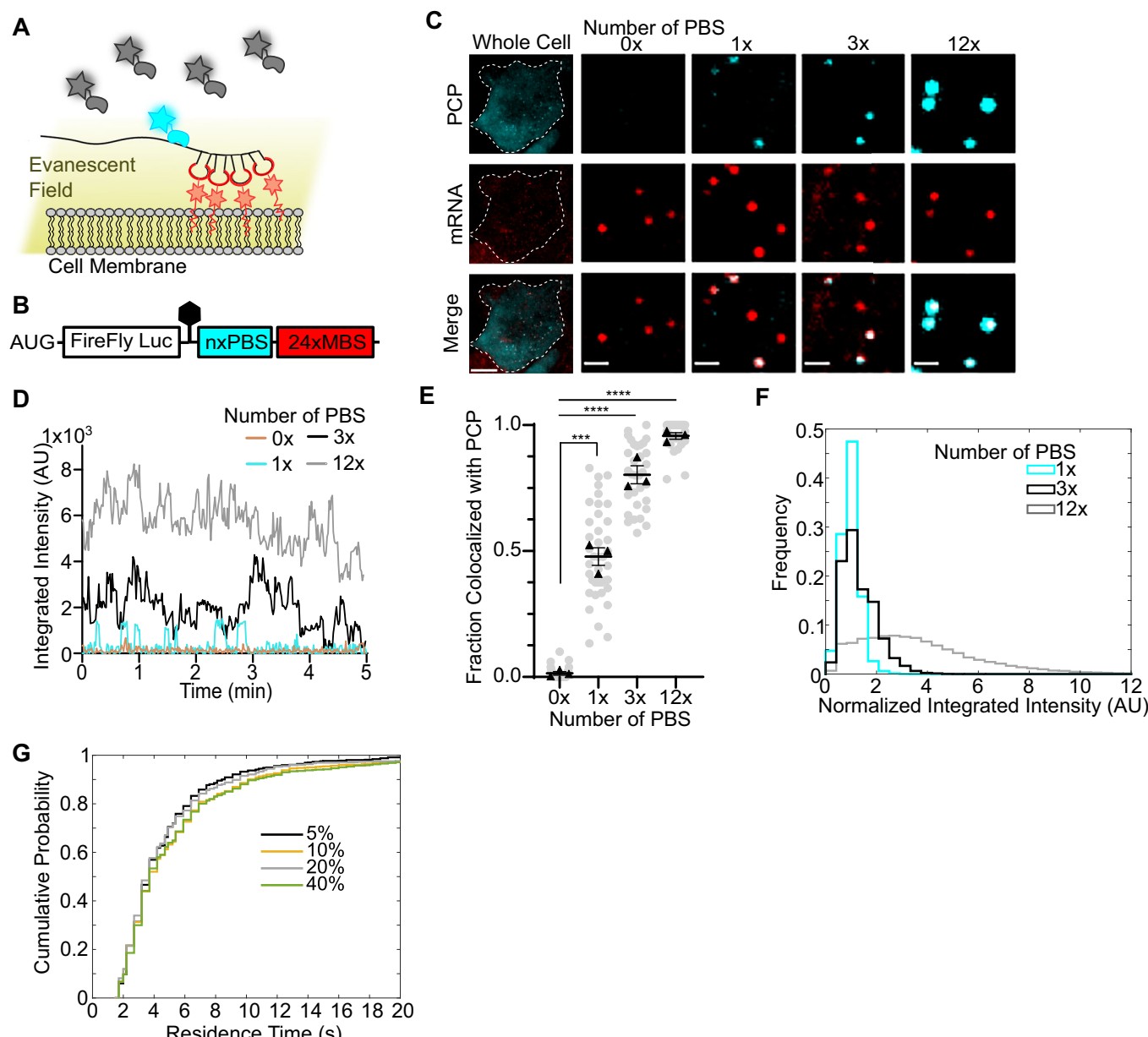

**Figure 1. Visualizing single proteins binding on mRNAs.**

(A) Scheme: tethering mRNAs to the plasma membrane allows visualizing single proteins bound to mRNAs by TIRFM. (B) The construct to validate single-molecule imaging of RBP–mRNA interactions. The mRNA encodes Firefly luciferase. In the 3′UTR, various numbers of PBS were inserted before 24xMBS. (C) Example of U-2 OS live-cell imaging of nxPBS-24xMBS reporters ($n = 0, 1, 3, 12$, respectively) scale bar = 1 μm; whole cell image scale bar = 10 μm. Dashed lines represent cell boundaries. tdPCP-HaloTag was used as a model RBP. Red: mRNA; Cyan: tdPCP-HaloTag-JFX-646. Colocalized signal indicates protein bound to mRNA. (D) Example trace of tdPCP-HaloTag intensity over time for nxPBS mRNAs. (E) Fraction of mRNAs with colocalized tdPCP-HaloTag at least once in 5 min (calculated only for cells with more than 15 mRNAs). Each dot represents one cell; bars represent mean ± SEM from three biological replicates. Black triangles represent the means of biological replicates. (0×: 12, 18, 10 cells; 1×: 10, 16, 13 cells; 3×: 10, 10, 10 cells; 12×: 12, 18, 9 cells). Statistical analysis using two-tailed, equal variance, $t$ test: 0× vs 1×, ***$P = 0.0002$; 0× vs 3× ****$P = 2.64E-05$, 0× vs 12× ****$P = 3.44E-07$). (F) Normalized integrated intensity of tdPCP-HaloTag spots localized to mRNA. Cyan: 1× PBS; Black: 3× PBS; Gray: 12× PBS. Data were compiled from three independent biological replicates. $P$ value for comparison of 1× to 3×: <0.0001; $P$ value for comparison of 1× to 12×: <0.0001; $P$ values calculated by Kolmogorov–Smirnov test. (1×: 148, 213, 193 3×: 229, 266, 270 12×: 212, 404, 194 tracks). (G) Cumulative distribution function (CDF) of "on" times of tdPCP-HaloTag localized to 1× PBS mRNA with varying laser power. Data were compiled from two independent experiments. $P$ values for comparison of 5 to 40%: 0.0335; 10% to 40%: 0.6229; 20% to 40%: 0.0463; $P$ values calculated by Kolmogorov–Smirnov test. (5%: 150, 191; 10%: 162, 118; 20%: 183, 155; 40%:145, 188 tracks). Source data are available online for this figure.

encodes firefly luciferase. We expressed the reporters in U-2 Osteosarcoma (U-2 OS) cell lines stably expressing tdMCP-mScarlet-CAAX and tdPCP-HaloTag. Using TIRFM, we visualized tdPCP-HaloTag that colocalized with MS2-labeled mRNA (Fig. 1C).

We tracked the mRNA and RBP for 5 min (Fig. 1D). We defined a bound state for an mRNA as tdPCP-HaloTag colocalization lasting at least eight seconds. There were essentially no detectable colocalized spots on samples with 0× PBS mRNA. By contrast, for samples with 1×, 3×, and 12× PBS mRNAs, we observed the coincidence of signal, indicating at least one bound protein, 50%, 80% and 95% of the time, respectively (Fig. 1E). We segmented the tdPCP-HaloTag intensity traces of single mRNAs into bound and unbound states. The integrated intensity for 1× PBS displayed a single peak. We normalized the PBS intensity using the mean of 1× PBS one. The normalized intensities of 3× and 12× PBS are higher, with distributions skewed to the right, as expected ($P$ value = <0.0001) (Fig. 1F). The intensity of 3× PBS and 12× PBS is not always 3 or 12 times that of a single tdPCP-HaloTag. This may be due to the unsaturated binding of tdPCP to PBS or the incomplete labeling of the RBP under our experimental conditions. In addition, for 1× PBS mRNAs, non-colocalizing tracks did not show any binding, while many colocalizing tracks exhibited multiple binding events. Therefore, the mRNAs might also contain misfolded PBS that cannot bind PCP. Despite the complexity of 3× and 12× PBS, the measured intensity of 1× PBS can be used as a stoichiometric benchmark for calibrating protein–mRNA interactions.

Next, we analyzed the binding kinetics of tdPCP-HaloTag to 1× PBS mRNA. We observed a characteristic on-and-off binding pattern. Because the "on" time relies on intrinsic binding affinity, whereas the "off" time depends on the unknown tdPCP concentration, we focused only on the "on" (residence) time. The distribution decays exponentially with an average of 3.6 s (Fig. 1G). To ensure that the observed binding events were not artifacts of dye photophysics or limited by photobleaching, we varied the illumination laser powers by up to eightfold. The residence time distribution remained unchanged (Fig. 1G), confirming that the measurement is independent of photophysics. To further validate our approach, we introduced a single G- > U mutation in the loop region of PBS (L4 mutant), which should reduce binding affinity by an order of magnitude (PBS WT: AUAUGG, $K_d$ = 1 nM; L4 mutant: AUAGGU, $K_d$ = 13 nM) (Appendix Fig. S1A) (Lim, 2002). We imaged tdPCP-HaloTag binding to both wild-type and L4 mutant PBS. To capture the transient binding of tdPCP to L4 mutant, we needed to use a faster acquisition time of 100 ms. As expected, the tdPCP-HaloTag binding duration was significantly shorter for the L4 mutant than for the wild-type PBS (Appendix Fig. S1B,C; Movies EV1 and EV2). This data further validates our method's sensitivity to single-molecule binding kinetics in live cells.

### Tagging endogenous ZNF598 with HaloTag

Building on this methodology, we next aimed to explore the interaction between the essential RQC factor ZNF598 and various mRNAs. Tagging ZNF598 at its endogenous locus is crucial because altering ZNF598 levels severely impacts RQC kinetics (Goldman et al, 2021). We employed CRISPR/Cas9 to endogenously tag ZNF598 with HaloTag at its C-terminus in U-2 OS cells (Fig. 2A) and obtained single clonal isolates ("Methods"). To validate the

ZNF598 insertion, we amplified genomic DNA at the target region by polymerase chain reaction (PCR), yielding a single PCR product of the expected size (Fig. 2B). Further analysis via DNA sequencing and western blot validated the homozygous insertion of the HaloTag (Fig. 2C; Appendix Fig. S2A). Staining the knock-in cell line with JFX-646 fluorescent dye revealed the cytoplasmic localization of ZNF598-HaloTag, consistent with previous reports (Appendix Fig. S2B) (Matsuo et al, 2017).

To validate that ZNF598-HaloTag is functional, we employed the SINAPS RQC reporter as previously described (Fig. 2D) (Wu et al, 2016; Yan et al, 2016; Morisaki et al, 2016; Pichon et al, 2016; Wang et al, 2016a; Goldman et al, 2021). Briefly, in this system, repetitive GCN4 peptide epitopes (SunTag) are placed at the N-terminus of the ORF. Upon translation, a single-chain variable fragment antibody against GCN4 fused to a super-folder GFP (scFv-sfGFP) binds the nascent peptide, allowing for tracking of single mRNA translation dynamics (Fig. 2E). An auxin-inducible degron (AID) at the N-terminus degrades mature protein once it is released from the ribosome, reducing background fluorescence (Fig. 2D). In addition, 24xMBS were inserted in the 3′UTR for visualizing mRNA via tdMCP-mScarlet-CAAX. Two different SINAPS reporters were developed, one carrying a polyadenosine sequence (poly(A)60) in the ORF prior to the stop codon and the other with "no-insert", serving as the "normal" mRNA control.

To validate the functionality of fluorescently tagged ZNF598, we transiently expressed the SINAPS reporters (poly(A)60 and no-insert) in the knock-in and wild-type cells and conducted ribosome runoff experiments. Runoff experiments use Harringtonine to effectively inhibit translation initiation by blocking the first peptide bond formation while allowing actively elongating ribosomes to "run off" the mRNA (Fig. 2E) (Fresno et al, 1977). After Harringtonine treatment, the survival probability of translation sites (TLS) over time reports on the speed at which existing ribosomes elongate through the ORF and terminate or are cleared from the mRNA. As expected, the TLS from the poly(A)60 reporters persisted longer than the no-insert one, consistent with previous findings (Goldman et al, 2021). Importantly, the survival probabilities of both poly(A)60 and no-insert control mRNAs in ZNF598-HaloTag knock-in cells were identical to those observed in wild-type cells (Fig. 2F,G; Movies EV3 and EV4). Given that reduced ZNF598 levels are known to compromise ribosome clearance from poly(A)60 mRNAs, this experiment demonstrates that endogenously HaloTag-labeled ZNF598 is functional.

### Visualizing recruitment of endogenous ZNF598 to translating mRNAs in live cells

As described above, we labeled endogenous ZNF598-HaloTag with JFX-646 and performed three-color imaging to simultaneously visualize ZNF598, the SINAPS reporter mRNAs, and their translation in live cells (Fig. 3A,B; Movies EV5 and EV6). We tracked the mRNA, TLS and ZNF598 for 5 min (Fig. 3C,D). Many ZNF598 spots did not colocalize with SINAPS mRNAs. Since the reporter mRNAs account for only a minority of the transcripts in the cell, we reason that ZNF598 is recruited to other "problematic" mRNAs in the cell. We quantified the fraction of SINAPS mRNAs colocalized with ZNF598. For non-translating mRNAs, there was no recruitment of ZNF598, as expected, given that ZNF598 interacts with ribosomes, not with mRNAs per se. Importantly,

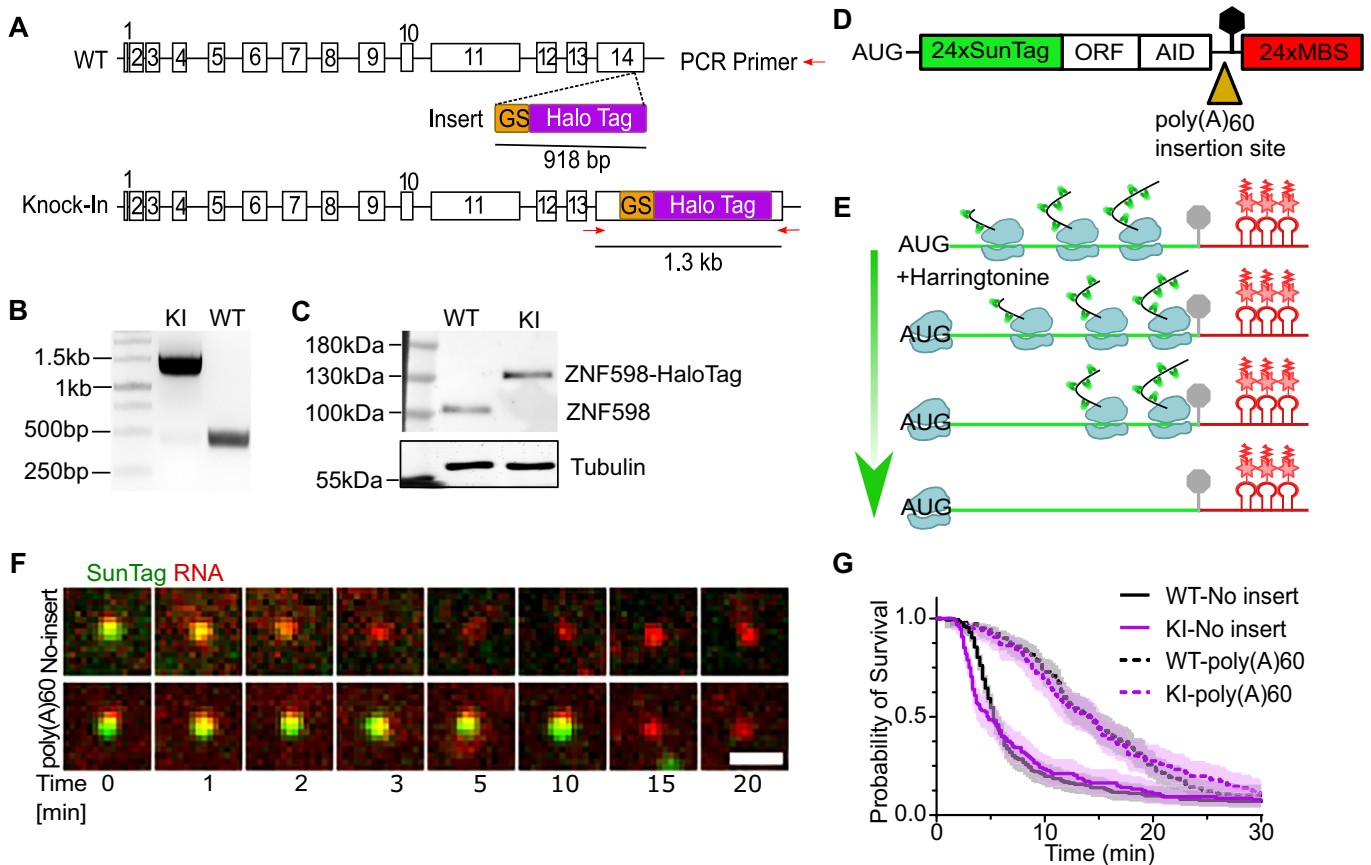

**Figure 2. Tagging endogenous ZNF598 with HaloTag in U-2 OS cells using CRISPR technology.**

(A) Scheme of knocking HaloTag into the endogenous *Znf598* locus in exon 14. Red arrows: PCR primers used to check the knock-in allele; GS: Glycine-Serine linker. (B) Genomic PCR amplicons of wild-type (WT) and Knock-In (KI) cell lines with primers shown in (A). (C) Western blot of wild-type (WT) and knock-in (KI) U-2 OS cells with ZNF598 antibody. Tubulin: loading control. (D) Diagram of the SINAPS reporter for single-molecule imaging of translation. ORF open reading frame, AID Auxin-inducible degron, MBS MS2 binding site. (E) Scheme of ribosome runoff assay. Harringtonine stops translation initiation while allowing elongating ribosomes to "runoff" from the mRNA. (F) Snapshots from Supplementary Movies 1 and 2 of ribosome runoff experiment in knock-in U-2 OS cells. The time reported was after adding harringtonine. Scale bar: 1 μm. (G) The cumulative survival probabilities of translation sites post-harringtonine treatment. The ribosome runoff experiments of no-insert and poly(A)60 reporters were conducted in the ZNF598 WT and KI cells, respectively. Shaded area: 95% confidence bounds computed using Greenwood's formula. WT-No insert: 6 cells, 172 mRNAs; WT-poly(A)60: 8 cells, 140 mRNAs; KI-No insert: 7 cells, 128 mRNAs; KI-poly(A)60: 6 cells, 116 mRNAs Data gathered from three biological replicates. Source data are available online for this figure.

there was a significantly higher fraction of translating poly(A)60 mRNAs associated with ZNF598 (~0.8) than with the no-insert control mRNA (~0.1) (*P* value = 0.0078) (Fig. 3E).

We next analyzed the number of ZNF598 molecules on SINAPS mRNAs. Although the importance of ZNF598 in RQC has been well established, it remains unclear whether ZNF598 specifically identifies the leading collided ribosomes or any of those in a queue of collisions. These two models would predict different numbers of molecules being recruited to long queues of ribosomes on the poly(A) reporter (Goldman et al, 2021). To distinguish between these two models, we quantified the integrated intensity of ZNF598-HaloTag on translating mRNAs and compared it with the tdPCP-HaloTag signal on the 1×, 3×, and 12× PBS controls (Fig. 3F; Appendix Fig. S3A). The normalized intensity of ZNF598 on the poly(A)60 reporters was significantly higher than on the 1× PBS mRNA (*P* value = <0.0001), and was comparable to or slightly higher than on the 3× PBS (*P* value = <0.0001). These data indicate that multiple ZNF598 molecules can be recruited onto a single

mRNA and support the model that ZNF598 recognizes collided ribosomes, regardless of whether they are in the leading position in the queue or not. We also analyzed the time spent by ZNF598 on poly(A)60 mRNAs. Some ZNF598 tracks persist for the whole 5 min, while others are transient, with an average duration of on average 3.5 min (Appendix Fig. S3B). However, since there is more than one ZNF598 on an mRNA, we cannot infer from this the residence time of a single ZNF598 on collided ribosomes.

## ZNF598 is a rate-limiting factor in RQC

Ribosome disassembly in RQC is a complex multistep process. After ZNF598 ubiquitinates the collided ribosomes, ASCC3 is recruited, likely to resolve the queue by disassembling the leading ribosome from the 3′ end (Juszkiewicz et al, 2020; Matsuo et al, 2020). Although the disassembly process has been reconstituted in vitro, it is unknown which step is rate-limiting in cells. Previously, we found that there were on average more than 40

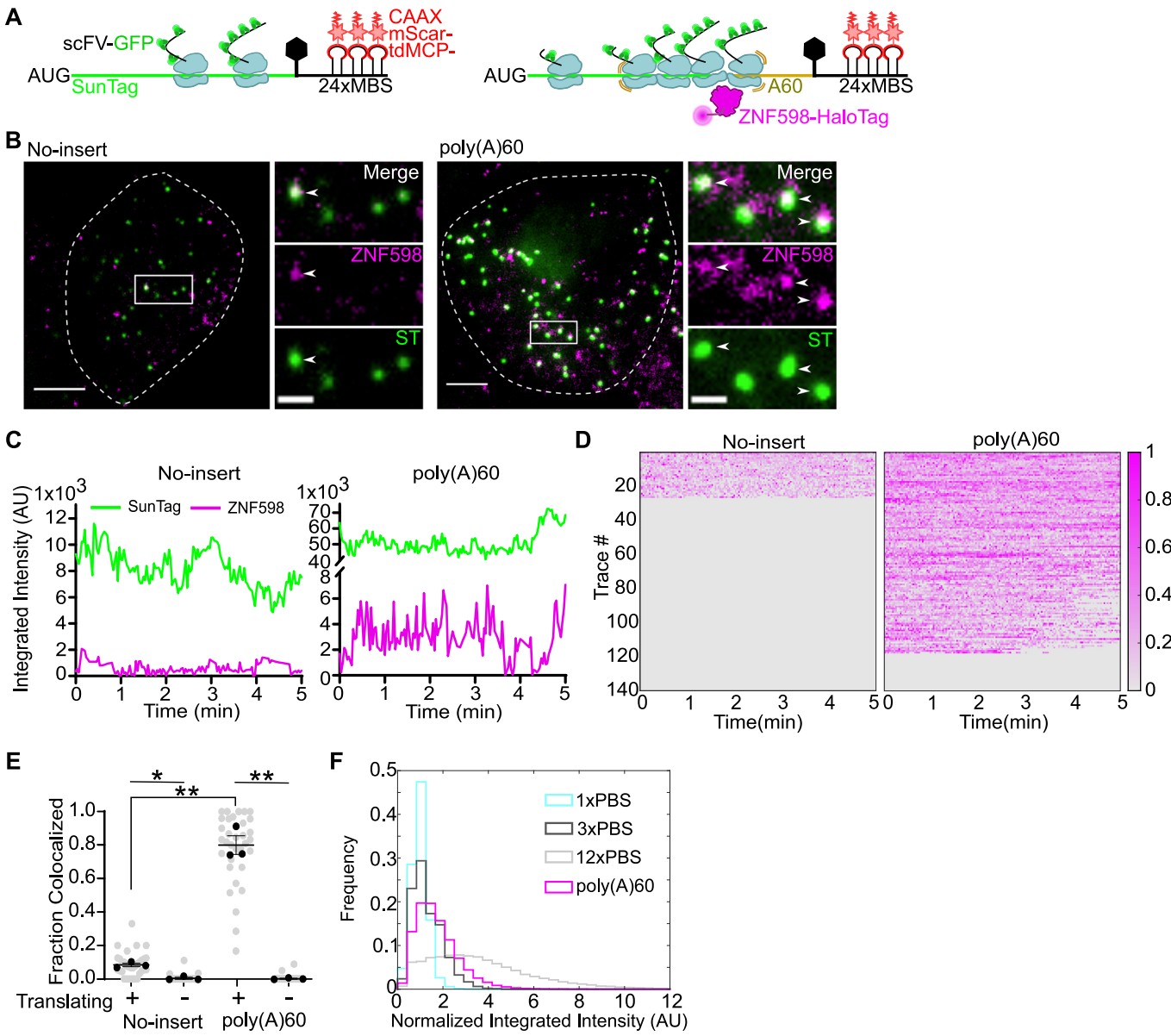

**Figure 3. Visualizing endogenous ZNF598 interacting with translating mRNA in live cells.**

(A) Scheme of the no-insert (left) and poly(A)60 (right) reporters with translating ribosomes. ZNF598-HaloTag (magenta) interacts with colliding ribosomes on poly(A)60 reporter. (B) Snapshots from live-cell imaging of no-insert (left) and poly(A)60 (right) reporters (Supplementary Movies 3 and 4). Green: translation sites; magenta: ZNF598-HaloTag; white arrowheads: ZNF598-HaloTag colocalized with translating mRNA. Dashed line represents cell boundaries. Scale bar: 5 μm and 2 μm for zoom-in. (C) Sample intensity traces of translation site and ZNF598-HaloTag over time for no-insert (left) and poly(A)60 (right) reporters. (D) Composite of normalized ZNF598 intensity trajectories on translating mRNAs for no-insert (left) and poly(A)60 (right) reporters. no insert: 10 cells, 140 tracks; poly(A)60: 8 cells, 140 tracks. (E) Fraction of mRNAs colocalized with ZNF598 at least once in 5 min (calculated only for cells with more than 15 mRNAs). Each dot represents one cell; bars represent mean ± SEM from three biological replicates. Black circles represent means of biological replicates. (no insert: 9, 7, 16 cells; poly(A)60: 9, 6, 20 cells;). Statistical analysis using two-tailed, equal variance, $t$ test: no insert-translating vs non-translating, **$P = 0.0058$; poly(A)60 translating vs non-translating, **$P = 0.0044$, no-insert vs poly(A)60, **$P = 0.0078$. (F) Normalized integrated intensities of ZNF598-HaloTag colocalized to translating mRNA. The histogram of 1× (cyan), 3× (black) or 12× PBS (gray) mRNAs from Fig. 1F were reproduced here for comparison. Data were compiled from two independent biological replicates. $P$ value for comparison of 1× to poly(A)60: <0.0001. $P$ value calculated by Kolmogorov–Smirnov test. (1×: 148, 213, 193; 3×: 229, 266, 270; 12×: 212, 404, 194; poly(A)60: 224, 147 tracks). Source data are available online for this figure.

ribosomes on the poly(A)60 mRNAs (Goldman et al, 2021). As established here, the measured stoichiometry of ZNF598 recruitment to mRNA is larger than one. So ZNF598 must target the trailing ribosomes as well. However, stoichiometry is far from saturating the potential binding sites (compared with 12× PBS, Fig. 3F). It was estimated that the concentration of ZNF598 is 50-fold lower than that

of ribosomes in HeLa cells (Kulak et al, 2014; Itzhak et al, 2016). One plausible explanation is that there is not enough ZNF598, and it is limiting the clearance of the poly(A)60 reporter. To test this hypothesis, we stably overexpressed ZNF598-HaloTag in wild-type U-2 OS cells containing SINAPS reporters (Fig. 4A,B) and performed ribosome runoff experiments (Fig. 2B). Indeed, the ribosome runoff

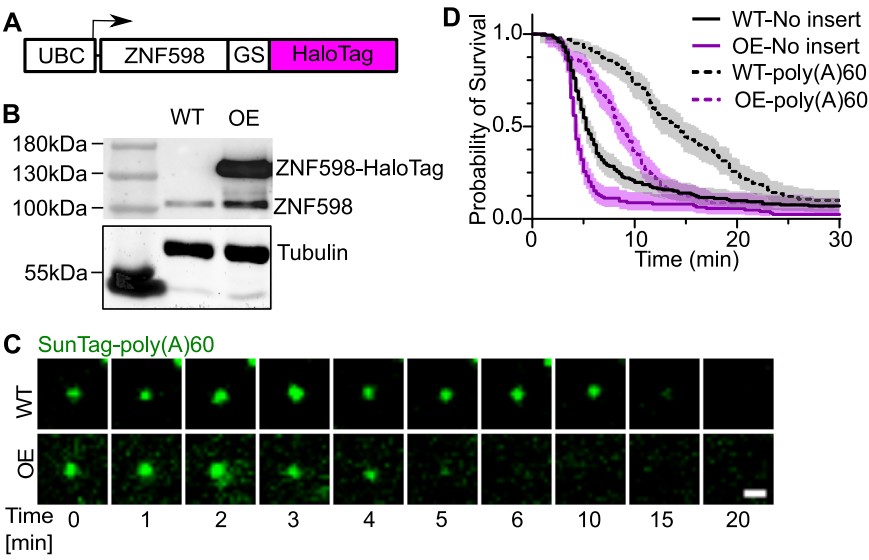

**Figure 4. ZNF598 is a rate-limiting factor of RQC.**

(A) The overexpression construct of ZNF598-HaloTag, which is identical to the endogenous, labeled one in Fig. 2. UBC ubiquitin C promoter, GS Glycine-Serine Linker. (B) Western blot of wild-type (WT) and overexpression (OE) U-2 OS cells with ZNF598 antibody. Tubulin: loading control. (C) Snapshots from Supplementary Movies 5 and 6 of ribosome runoff experiment in WT and OE U-2- OS cells. The time reported was after adding harringtonine. Scale bar: 1 µm. (D) The cumulative survival probability of translation sites on mRNA post-harringtonine treatment. The ribosome runoff experiments of no-insert and poly(A)60 reporters were conducted in the ZNF598 WT and OE cells, respectively. Shaded area: 95% confidence bounds computed using Greenwood's formula. 6–9 cells; 140–204 mRNAs were used for each condition. WT-No insert: 6 cells, 172 mRNAs; WT-poly(A)60: 8 cells, 140 mRNAs; OE-No insert: 7 cells, 125 mRNAs; OE-poly(A)60: 7 cells, 138 mRNAs. Data gathered from three biological replicates. Source data are available online for this figure.

for poly(A)60 mRNA was 1.5 times faster in cells overexpressing ZNF598 than in wild-type cells. Importantly, ribosome runoff from the no-insert control mRNA was also slightly faster (Fig. 4C,D; Movies EV7 and EV8). This is not due to fewer ribosomes in the queue, because the number of nascent peptides (NAPs) on the poly(A)60 reporter in ZNF598 overexpression cells is statistically the same as in the wild-type cells (P value = 0.2719) (Appendix Fig. S4A,B). These results support a model wherein ZNF598 is rate-limiting in the RQC process in U-2 OS cells.

## A fraction of "normal messages" are targets of RQC and resistant to harringtonine treatment

While 80% of poly(A) mRNAs were associated with ZNF598, we were surprised that about 10% of translating "normal" no-insert mRNAs also colocalize with ZNF598 (Fig. 3E). Given that these are "normal" transcripts, their recognition by ZNF598 was unexpected. There are at least two models to explain this interaction. First, transient collisions likely occur with some frequency on all messages, and ZNF598 may randomly sample these colliding ribosomes. Second, some "normal" mRNAs might be damaged, resulting in authentic ribosome collisions at the damage site, which recruit ZNF598. These damaged sites would likely escape detection by traditional sequencing or ribosome profiling experiments because they are random and will not be enriched. To distinguish between these two models, we imaged ZNF598 recruitment to no-insert mRNAs during ribosome runoff experiments. We found that mRNAs with pre-bound ZNF598 have longer runoff time compared with the ones without ZNF598 (Appendix Fig. S5). However, this assay was very low throughput, because each cell

only has a few harringtonine-resistant mRNAs. Therefore, we performed a ribosome runoff assay on no-insert reporters and only quantified ZNF598 recruitment at 10 min after harringtonine treatment to accumulate more statistics (Fig. 5A). Approximately 90% of translating mRNAs had ribosomes runoff completely within 10 min after harringtonine treatment, while 10% were resistant to harringtonine runoff (Fig. 5B; Movies EV9 and EV10). We quantified the recruitment of ZNF598 to the same set of translating mRNAs. Before treatment, about 10% of translating mRNAs were associated with ZNF598, and this number increased to ~50% 10 min after treatment (Fig. 5C,D). These data suggest that some normal mRNAs are resistant to harringtonine runoff because they are damaged, occupied by collided ribosomes, and recognized by ZNF598. The nature of mRNA damage may be multifaceted and will not easily be defined by our approaches.

## ZNF598 is a limiting factor for responding to globally damaged transcripts

UV light induces the formation of pyrimidine dimers on mRNAs and potently induces ribotoxic stress through collided ribosomes (Wurtmann and Wolin, 2009; Wu et al, 2020; Sinha et al, 2024; Iordanov et al, 1998). We observed ZNF598 on a fraction of "normal" reporter transcripts, suggesting they may be damaged. To further investigate, we applied UV light to induce mRNA damage and observe ZNF598 recruitment. We hypothesized that inducing UV damage in cells expressing the "normal" no-insert mRNAs would increase ZNF598 recruitment to those mRNAs. We imaged the ZNF598-HaloTag knock-in cell line before and after UV light exposure (Fig. 6A,B; Movies EV11 and EV12). Surprisingly, there

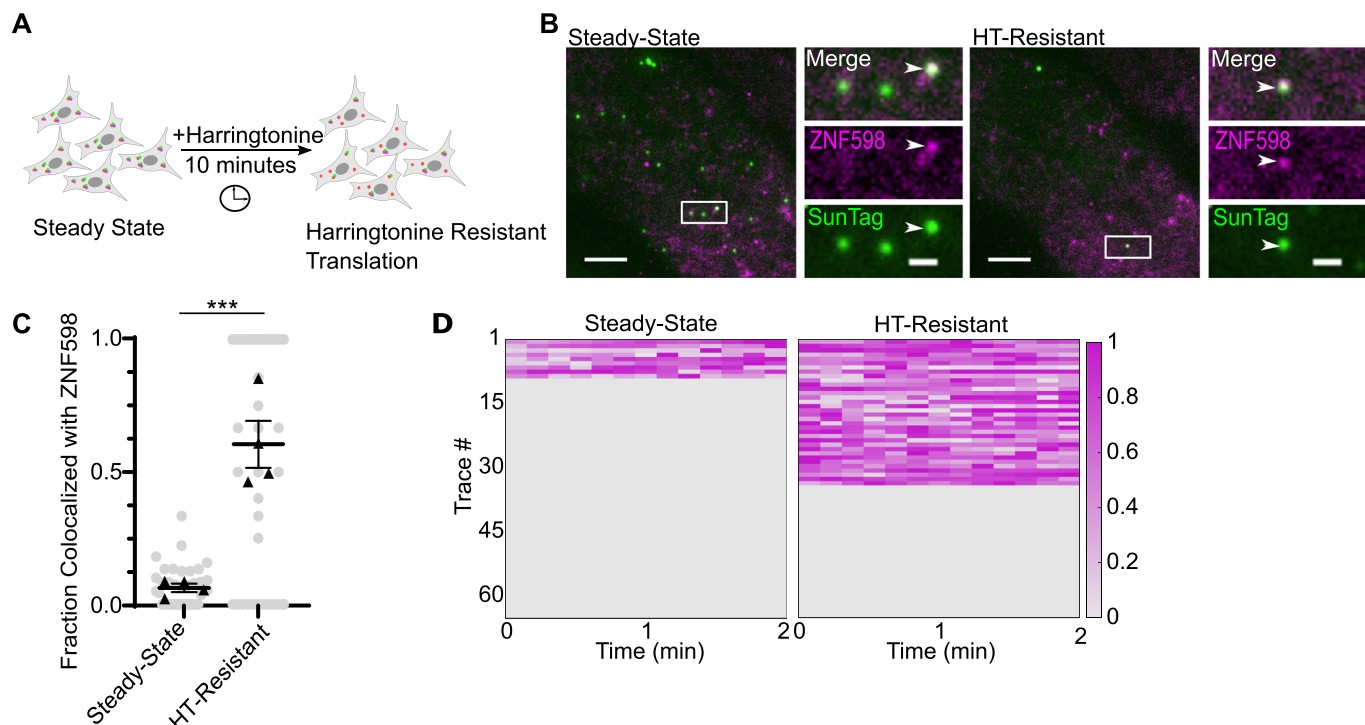

**Figure 5. Harringtonine-resistant translation sites are recognized by ZNF598.**

(A) Scheme of ZNF598 recruitment to no-insert mRNAs after harringtonine treatment. (B) Snapshots from live-cell imaging at steady state (left) and post-harringtonine (right) (Supplementary Movies 7 and 8). Green: translation sites; magenta: ZNF598-HaloTag; white arrowheads: ZNF598 colocalized with translating mRNA. Scale bar: 5 µm and 2 µm for zoom-in. (C) Fraction of mRNAs colocalized with ZNF598 at least once within 2 min (calculated only for cells with translating mRNA). Each dot represents one cell; bars: mean ± SEM from four biological replicates. Black triangles: the means of biological replicates. (Steady state: 15, 9, 5, 7 cells; HT-Resistant: 15, 9, 5, 7 cells). Statistical analysis using two-tailed, equal variance, *t* test: Steady-state vs HT-resistant, ***$P = 0.009$). (D) Composite of normalized ZNF598-HaloTag intensity at steady state (left) and after harringtonine treatment (right). Source data are available online for this figure.

was no statistically significant difference in ZNF598 recruitment to the no-insert mRNAs after UV exposure (Fig. 6C; Appendix Fig. S6A,B; Movie EV11). This result was puzzling as ribosome runoff on no-insert mRNAs was dramatically slowed after UV light exposure, suggesting the existence of heavy mRNA damage (Appendix Fig. S6C,D; Movie EV13). To reconcile these findings, we hypothesized that ZNF598 became even more rate-limiting under these conditions, as it would now be distributed across the entire damaged transcriptome. To test this prediction, we examined ZNF598 recruitment to poly(A)60 mRNA after UV damage. Remarkably, there was a dramatic reduction in ZNF598 recruitment to the poly(A)60 mRNAs after UV damage, similar to the level seen on the no-insert control (Fig. 6C,D; Movie EV12), even though the number of NAPs on poly(A)60 mRNAs increased after UV damage (Mean: $-$UV = 40.3; $+$UV = 48.1; $P$ value < 0.001) (Appendix Fig. S6E,F). These data further support that ZNF598 is a limiting factor for RQC. During global cellular insult and massive ribosomal collisions, ZNF598 cannot adequately respond to even the aberrant reporter RQC mRNAs.

## Discussion

Visualizing single-protein factors on specific mRNAs and observing their functions has been challenging. Building on a single-molecule

translation assay, we tethered mRNAs to the plasma membrane and used TIRFM to visualize and correlate single regulatory protein factors with translation output. Our study demonstrated the feasibility of this approach using well-characterized PCP-PBS interactions on MS2-labeled mRNAs. By varying the number of PBS on reporter mRNAs, we established a calibration standard of protein–RNA interactions. Applying this method, we investigated the interaction between the endogenous RQC factor ZNF598 and mRNAs. ZNF598 interacts preferentially with an RQC mRNA substrate with stalling sequences. Intensity measurements revealed that multiple ZNF598 molecules bind to a single translating RQC mRNA, suggesting that ZNF598 can associate with multiple stalled ribosomes within a queue, not just the leading collided ribosome. Our results showed that ZNF598 is a limiting factor in the RQC process in cells, as its overexpression can speed up ribosome clearance. Interestingly, ZNF598 occasionally binds to supposedly "normal" messages, which we propose to be somehow damaged mRNAs. Importantly, this latter observation suggests that ZNF598 plays a broader role in quality control beyond just targeting well-characterized RQC substrates. Under global mRNA damage by UV light, the limited availability of ZNF598 becomes even more apparent, as it is insufficient to manage the increased load of RQC substrates. By developing and applying the method to visualize protein–mRNA interactions and track their downstream consequence, our study has revealed previously unseen dynamics in the RQC pathway.

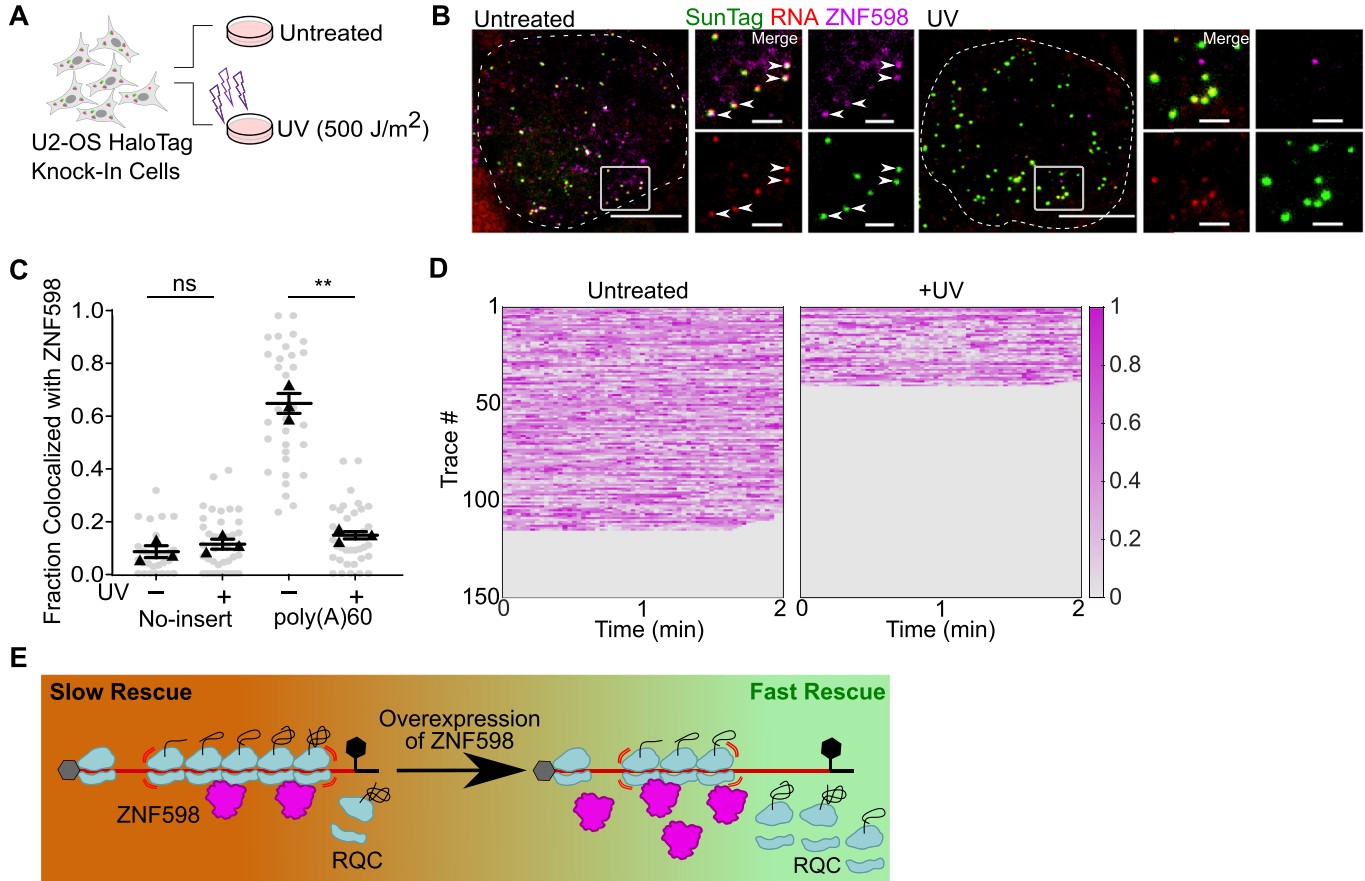

**Figure 6. ZNF598 is a limiting factor for global UV damage.**

(A) Scheme of UV damage assay. (B) Snapshot of live-cell imaging at steady state (left) and post-UV damage (right) (Supplementary Movies 9 and 10). Green: translation sites; magenta: ZNF598-HaloTag; white arrowheads: ZNF598 colocalized with translating mRNA. The dashed line represents cell boundaries. Scale bar: 5 µm and 2 µm for zoom-in. (C) Fraction of mRNAs colocalized with ZNF598 at least once in 2 min (calculated only for cells with translating mRNA). Each dot represents one cell; bars: mean ± SEM from three biological replicates; black triangles: means of biological replicates. (No insert/-UV: 12, 8, 11 cells; No insert/+UV: 14, 13, 12; poly(A)60/-UV: 10, 10, 12 cells; poly(A) 60/ + UV: 12, 11, 16 cells). Statistical analysis using two-tailed, equal variance, *t* test: ST/-UV vs ST/ + UV, ns=not significant, *P* = 0.138, poly(A)60/-UV vs poly(A)60/ + UV **P* = 0.0048). (D) Composite normalized ZNF598-HaloTag intensity at steady state (left) and after UV damage (right) for poly(A)60 mRNAs. (−UV: 10 cells, 150 tracks; +UV: 13 cells, 150 tracks). (E) Model for ribosome rescue indicating that ZNF598 is a rate-limiting factor in the RQC pathway. Source data are available online for this figure.

We observed that multiple ZNF598 molecules could bind to a single RQC mRNA, suggesting that any collided ribosome within a queue can be recognized by ZNF598. At present, no high-resolution structure shows ZNF598 bound to collided ribosomes. Nevertheless, our results fit well with earlier biochemical evidence suggesting that ZNF598 may act on the trailing collided ribosomes. The proportion of eS10 that becomes ubiquitinated—the ZNF598 signature target—increases with the length of the ribosome queue (Juszkiewicz et al, 2018). In addition, UV-cross-linking experiments have detected ZNF598 on mRNAs at multiple sites upstream of the stall sequence (Garzia et al, 2017). Although more than one ZNF598 is present, the measured number is far from saturating available binding sites. Several factors could explain these observations. First, the concentration of ZNF598 is low and cannot saturate the available binding sites. Previous studies have shown that ZNF598 is ~50 times less abundant than ribosomes (Kulak et al, 2014; Itzhak et al, 2016). As we have shown, normal messages may also be damaged in cells, creating a higher-than-anticipated

demand for ZNF598. Second, the collided ribosomes may form a trisome (Juszkiewicz et al, 2018; Ikeuchi et al, 2019; Matsuo et al, 2020; Narita et al, 2022; Best et al, 2023). In cells, trisomes may even be the preferred structures for RQC to occur (Juszkiewicz et al, 2018; Matsuo et al, 2020, 2023; Madern et al, 2025). Formation of trisomes may reduce the number of binding surfaces for ZNF598 compared with disomes. Third, the enzymatic activity of ZNF598 may not require long-term binding. Finally, our experiment may underestimate the stoichiometry due to technical limitations. For example, although we have chosen the most efficient labeling condition of ZNF598, there might still be a fraction of unlabeled ZNF598 in the cell. Uneven TIRF excitation may lead to underestimation if the molecule is deeper in the evanescent field. Nonetheless, the main conclusion is clear: more than one ZNF598 molecule is associated with a single RQC mRNA, implying that the trailing collided ribosomes can also be targeted by ZNF598. This study also showed that overexpressing ZNF598 speeds up ribosomal clearance on poly(A) mRNAs, suggesting that ZNF598-

mediated ubiquitination, rather than ribosome splitting, is a rate-limiting step in RQC (Fig. 6E).

Stochastic variation in ribosome elongation rates makes ribosome collisions inevitable even on normal transcripts. How cells differentiate between transient and problematic ribosome collisions remains unclear. We demonstrated that ZNF598 is a limiting factor in RQC. Why do cells not express higher levels of ZNF598 to increase RQC efficiency? We hypothesize that maintaining a low concentration of ZNF598 is crucial for discriminating between these types of collisions. At low concentrations, ZNF598 may take minutes to ubiquitinate collided ribosomes, allowing transient pauses to resolve on their own. However, once a ribosome is tagged by ZNF598, its clearance of ribosomes occurs rapidly (within seconds). Conversely, if ZNF598 concentrations are too high, it might inappropriately target physiological transient pausing. A recent study reported ribosome clearance time of up to 20 min from the initial collision to resolution (Madern et al, 2025), much slower than our previous measurement of ~10 s, likely due to differences in experimental conditions (Goldman et al, 2021). In a long ribosome queue, the rate-limiting ubiquitination step may be alleviated when trailing ribosomes are targeted by ZNF598 while they are waiting, as we have shown. Our findings, in conjunction with existing literature, suggest the rate-limiting nature of ZNF598-mediated ubiquitination ensures that persistent collisions in long queues are preferentially targeted by RQC.

A striking observation from this study is that a small fraction of "normal" mRNAs (~10%) was targeted by ZNF598. After 10 min of harringtonine treatment, 50% of the persistently "translating" mRNAs were associated with ZNF598, suggesting that these are damaged transcripts containing stalled ribosomes. If we generalize to the entire transcriptome, this could imply that there is a significant burden for the RQC system. It has been similarly argued that a substantial percentage of nascent proteins is degraded immediately upon synthesis (Schubert et al, 2000; Vabulas and Hartl, 2005). Defective mRNAs would be a significant source of these nascent proteins destined for degradation. The presence of ZNF598 on these defective mRNAs underscores the critical role of RQC in maintaining cellular homeostasis and preventing the accumulation of faulty proteins.

UV radiation induces the formation of pyrimidine dimers, causing decoding defects as ribosomes stall on the damaged mRNA (Wurtmann and Wolin, 2009; Wu et al, 2020; Sinha et al, 2024). Recent studies have shown an increase in eS10 ubiquitination after UV radiation, a hallmark of ZNF598-targeted ribosome collisions(Wu et al, 2020; Sinha et al, 2024). However, excessive collision elicits ribotoxic stress response (RSR) and integrated stress response (ISR), both regulated by the kinase ZAK (Wu et al, 2020; Iordanov et al, 1998). An appealing model suggests that the abundance and duration of the collisions, sensed by ZAK, function as a molecular gauge to balance cell survival and death. Despite this, it remains unclear why the cellular RQC pathway fails to handle the excessive collision load. We initially hypothesized that UV treatment would increase ribosome collisions and consequently enhance ZNF598 recruitment. Surprisingly, UV exposure resulted in less ZNF598 recruitment to both the no-insert and poly(A)60 mRNA. In hindsight, this makes total sense—UV-induced transcriptome-wide RNA damage likely sequesters ZNF598 on other endogenous mRNAs. Under normal conditions, ZNF598 is already in limited supply, handling homeostatic levels of damaged

transcripts. When the entire transcriptome is damaged, the cellular RQC machinery, initiated by ZNF598, struggles to clear the collisions, leading to the activation of RSR and ISR. This highlights the delicate balance cells must maintain between robust management of homeostatic ribosome collision and activating stress response when the damage load becomes overwhelming.

Despite technical advancements, there are limitations to the method presented in this work. We used the MBS/MCP system to tether the mRNA to the membrane for TIRF illumination. Therefore, the method cannot be used to visualize the protein–mRNA interaction in nuclei or cytosolic membraneless organelles. In addition, the evanescent field decreases exponentially in the axial direction, which causes heterogeneous brightness in the TIRF field due to mRNA fluctuations. Consequently, it is difficult to determine the number of RBPs on an mRNA accurately. Although there are clear limitations, this technique is broadly applicable to studying protein–RNA interactions that are not significantly influenced by membrane tethering. In particular, we anticipate it will be effective for investigating many protein–mRNA interactions occurring in freely diffusing mRNAs within the cytoplasm. Furthermore, our approach to visualizing RBP recruitment to mRNA is limited to proteins that are not highly abundant within the cell because the background noise may be overwhelming. To study endogenous proteins, we used CRISPR-mediated tagging for the protein of interest. Despite the increasing accessibility of CRISPR technology, knocking in long tags remains challenging, and certain genes may be incompatible with tagging.

In summary, we developed a single-molecule assay that enables real-time tracking of single-protein–mRNA interactions and translation dynamics in live cells. Our finding provides strong evidence for ZNF598's critical role in RQC. This work merely scratches the surface of this method's potential. Future studies could explore additional factors to further elucidate the temporal dynamics of RQC. Moreover, this study opens the door for wide investigations into protein–mRNA interactions at the single-molecule level and their implications for cellular health and disease.

## Methods

**Reagents and tools table**

| Reagent/resource | Reference or source | Identifier or catalog number |
| --- | --- | --- |
| **Experimental models** | | |
| HEK293T cells | ATCC | RRID: CVCL_0063 |
| U-2 OS cells | ATCC | RRID: CVCL_0042 |
| U-2 OS cells expressing SunTag Reporters | Goldman et al, 2021 | N/A |
| **Recombinant DNA** | | |
| pUBC-Fluc-24xMS2 | This paper | |
| pUBC-Fluc-1xPP7-24xMS2 | This paper | |
| pUBC-Fluc-3xPP7-24xMS2 | This paper | |
| pUBC-Fluc-12xPP7-24xMS2 | This paper | |
| pUBC-Fluc-L41xPP7-24xMS2 | This paper | |

| Reagent/resource | Reference or source | Identifier or catalog number |
|---|---|---|
| pcDNA5-CMV-ST-24xMS2 | Goldman et al, 2021 | |
| pcDNA5-CMV-ST-poly(A)60-24xMS2 | Goldman et al, 2021 | |
| pUBC-scFV-sfGFP | Dr. Bin Wu | Addgene #84563 |
| pUBC-tdMCP-mScarlett-RFP-CAAX | This paper | |
| pUBC-ZNF598-HaloTag | This paper | |
| **Antibodies** | | |
| ZNF598 antibody | Thermo Fisher | Catalog# 703601 |
| Tubulin antibody | ProteinTech | Catalog# 10094-1-AP |
| Anti-GFP antibody | Aves Lab | GFP-1010; RRID:AB_2307313 |
| Goat anti-Chicken IgY (H + L) Secondary Antibody, Alexa Fluor 488 | Thermo Fisher Scientific | A-11039; RRID:AB_142924 |
| Anti-rabbit HRP secondary antibody | Santa Cruz Biotechnology | sc-2357; RRID:AB_628497 |
| **Oligonucleotides and other sequence-based reagents** | | |
| ZNF598 gRNA DNA template | This paper | Appendix Table S1 |
| tracrRNA | This paper | Appendix Table S1 |
| Genotyping FWD Primer ZNF598 | This paper | Appendix Table S2 |
| Genotyping REV Primer ZNF598 | This paper | Appendix Table S2 |
| Sequencing Primer REV Halo | This paper | Appendix Table S2 |
| FWD Amplification primer of sgRNA DNA | This paper | Appendix Table S2 |
| REV Amplification primer of sgRNA DNA | This paper | Appendix Table S2 |
| FWD Left Homology Arm-GS Linker-Primer for Donor DNA amplification | This paper | Appendix Table S2 |
| REV Right Homology Arm Primer for Donor DNA amplification | This paper | Appendix Table S2 |
| SunTag_v4-Cy3 smFISH probes | This paper | Appendix Table S3 |
| **Chemicals, enzymes, and other reagents** | | |
| Dulbecco's Modified Eagle Medium (DMEM) | Corning | Catalog# 10-013-CV |
| Fetal bovine serum | Millipore Sigma | F4135-500ML |
| Penicillin/streptomycin | Millipore Sigma | P0781 |
| PVDF filter | Millipore Sigma | SLHV013SL |
| nucleofection solution | Lonza | V4XC-1024 |
| Cas9 Electroporation Enhancer | IDT | Catalog# 1075915 |
| 4D-Nucleofector Core Unit | Lonza | AAF-1003X |
| fibronectin | Millipore Sigma | F1141-2MG |
| Platinum SuperFi II DNA polymerase | Thermo Fisher | Catalog# 12361010 |
| X-tremeGENE HP DNA transfection reagent | Millipore Sigma | Catalog# 6366244001 |

| Reagent/resource | Reference or source | Identifier or catalog number |
|---|---|---|
| Opti-MEM Reduced Serum Medium | Gibco | Catalog# 31985062 |
| JFX-646 Halo dye | Dr. Luke Lavis | |
| FluoroBrite DMEM | Gibco | A1896701 |
| 1xGlutaMAX | Gibco | Catalog# 35050061 |
| Paraformaldehyde | Electron Microscopy Sciences | Catalog# 50-980-492 |
| nuclease-free water | Quality Biological | 351-029-131CS |
| Dulbecco's Phosphate-Buffered Saline (DPBS) | Corning | 21-031-CV |
| tritonx100 | Millipore Sigma | T8787-100mL |
| ProLong Diamond antifade mounting reagent with DAPI | Invitrogen | P36962 |
| 3-Indoleacetic acid | Millipore Sigma | I2886 |
| Harringtonine | Cayman Chemical Company | Catalog# 15361 |
| Agarose | Invitrogen | Catalog# 16500100 |
| TAE Buffer | Quality Biological | Catalog# 351-008-131 |
| Ethidium Bromide | Invitrogen | Catalog# 15585011 |
| Protease Inhibitor | Millipore Sigma | P8340 |
| Hygromycin | InvivoGen | ant-hg-1 |
| BSA | VWR | VWRV0332-25G |
| SSC buffer | Corning | 46-020-CM |
| PBS buffer | Corning | 46-013-CM |
| Formamide | Millipore Sigma | F9037-100ML |
| *E. coli* tRNA | Millipore Sigma | 10109541001 |
| Ultrapure BSA | Invitrogen | AM2616 |
| Dextran sulfate | Millipore Sigma | D8906-100G |
| Salmon Sperm | Invitrogen | 15632011 V |
| Ribonucleoside vanadyl complex | NEB | S1402S |
| SUPERase In | Thermo Fisher | AM2694 |
| Amino 11-ddUTP | Lumiprobe | A5040 |
| Terminal deoxynucleotidyl transferase (TdT) | Thermo Fisher | EP0162 |
| Cy3-NHS ester | Lumiprobe | 41020 |
| Costar Spin-X centrifuge column | Corning | 8162 |
| Magnesium chloride | Millipore Sigma | M2670-500G |
| Sodium acetate | Millipore Sigma | S2889 |
| 18 mm # coverslips | Fisher | 12-545-100 |
| Frosted glass slides | Thermo Fisher | 12-552-3 |
| **Software** | | |
| u-track software | Jaqaman et al, 2008 | |

| Reagent/resource | Reference or source | Identifier or catalog number |
|---|---|---|
| AirLocalize Software | Lionnet et al, 2011 | |
| MatLab | MathWorks | |
| Live-Cell Track Analysis Software | This paper | |
| Tdetector (step detection algorithm) | Chen et al, 2014 | |
| ImageJ (version 2.3.0) | NIH | |
| FISH-Quant Software | Mueller et al, 2013 | |
| uLocalize smFISH-IF Analysis | This paper | |
| **Other** | | |
| 4D-Nucleofector X Kit S | Lonza | V4XC-2032 |
| NEB Q5 Site-Directed Mutagenesis Kit | NEB | E0554S |
| Monarch PCR and DNA clean-up kit | NEB | T1030L |
| T7 high-yield RNA synthesis kit | NEB | E2050 |
| DNA extraction kit | Qiagen | Catalog# 69504 |
| UV StrataLinker 2400 UV Crosslinker | Stratagene | |

## Cell lines and culture conditions

All cells were cultured in DMEM supplemented (Corning, 10-013-CV) with 10%FBS (Millipore Sigma, F4135-500ML) and 100 units penicillin and 0.1 mg/mL streptomycin (Millipore Sigma, P0781) and maintained at 37 °C and 5% $CO_2$. Cells were passaged every 2–3 days once they reached ~75% confluency.

## Generation of stable cell lines

Lentiviral particles were generated by transfecting HEK293T cells with either tdMCP-mscarlet-RFP-CAAX, PCP-HaloTag, ZNF598-Halo-Tag, scFv-sfGFP plasmids along with viral packaging accessory plasmids. Forty-eight hours following transfection, the viral supernatant was collected, spun down to remove cellular contents, and filtered through a 0.45-μm PVDF filter (Millipore SLHV013SL). The filtered supernatant was applied directly to U-2 OS cells (American Type Culture Collection HTB-96). Viral transduction was performed sequentially by first infecting U-2 OS cells with PCP-HaloTag or ZNF598-HaloTag and performing fluorescence-activated cell sorting (FACS) for positive cells. This positive population was then infected in the same manner with scFv-sfGFP and/or tdMCP-mScarlet-RFP-CAAX and sorted for expressing cells.

Knock-in cell line tagging endogenous ZNF598 with HaloTag were generated by electroporation of Cas9 RNP as described in (Liu et al, 2020). Briefly, sgRNA and tracrRNA (1:1 ratio) were annealed by mixing and heated to 95 °C for 5 min in thermocycler (see Appendix Table S1). The mixture was allowed to cool on benchtop for 5 min. The RNP complex was formed by mixing 10 μg/μL of Cas9 protein with 50 μM sg:tracr sgRNA at a ratio of 1:1.2 and allowed to incubate for 20 min at room temperature. ~1.6 × 10⁶ U-2 OS cells were trypsonized, centrifuged, and the cell pellet was resuspended in 2 mL of DPBS. Cells were centrifuged again, and the cell pellet was resuspended in 40 μL of nucleofection solution (Lonza V4XC-1024). In total, 20 μL of resuspended cells

were added to RNP solution. In all, 2 μg of Donor DNA and 1 μL Cas9 Electroporation Enhancer (IDT # 1075915) were added to RNP cell mixture. Electroporation was performed according to the manufacturer's instructions on the 4D-Nucleofector Core Unit (Lonza AAF-1003X). SE Cell Line 4D-Nucleofector X Kit S with code DN-100 was used. After electroporation, DMEM complete was added and cells were seeded in a six-well plate coated with fibronectin (Millipore Sigma, F1141-2MG) (1:100). Cells were then sorted for positive knock-in using FACS. For monoclonal isolates single cells with a positive HaloTag signal were seeded into a 96-well plate in complete DMEM. Media was replenished with fresh media every 2 days. Single colonies were allowed to grow until 70–80% confluency. Single monoclonal colonies were then transferred to a 24-well plate and allowed to reach 70–80% confluency. Once confluent, they were transferred to a six-well plate and used for further testing of correct knock-in.

## Plasmids

The nxPP7 reporters were derived by inserting either 1, 3 or 12 PBS into our original plasmid phage-UBC-Fluc-24xMS2. In total, 1× L4-PBS (Lim, 2002) was generated using NEB Q5 Site-Directed Mutagenesis Kit (NEB E0554S) on the phage-UBC-Fluc-1xPBS-24xMS2 plasmid.

The "no insert" and poly(A)60 SunTag reporters were acquired from Goldman et al, 2021. The SunTag reporters from this study were derived from our original SINAPs reporter with the modifications discussed in Goldman et al, 2021 aimed at reducing cryptic splice isoforms. This base construct, pcDNA5_CMV_ST, contains a NanoLuciferase, BFP, and AID downstream of the SunTag in the ORF (1254 codons).

To create pUBC-tdMCP-mScarlett-RFP-CAAX, we first removed HaloTag from phage-UBC-tdMCP-HaloTag-CAAX and inserted mScarlet-RFP. To create pUBC-ZNF598-HaloTag, we first eliminated the NLS-HA-tdMCP in pUBC-NLS-HA-tdMCP-HaloTag and inserted ZNF598 from pcDNA-ZNF598-TEV-3xFLAG to generate pUBC-ZNF598-HaloTag which we used to generate lentiviral particles for the overexpression of ZNF598. We used the pubc-OSTIR1-IRES-scFv-sfGFP-nls plasmid from our original publication.

## Generation of sgRNA

The DNA template of sgRNA was generated by PCR amplification of two overlapping primers consisting of the CRISPR RNA template and gRNA template containing a T7 promoter (see Appendix Table S2). PCR using Platinum SuperFi II DNA polymerase (Thermo Fisher 12361010) was performed according to the manufacturer's protocol. The sgRNA DNA template was collected using Monarch PCR and DNA clean-up kit (NEB T1030L) according to the manufacturer's protocol. In vitro transcription of sgRNA DNA template was performed using T7 high-yield RNA synthesis kit (NEB E2050) according to the manufacturer's protocol.

## Generation of donor DNA

Donor DNA was synthesized by PCR amplification of the HaloTag region from pUBC-ZNF598-HaloTag plasmid. Briefly, PCR amplification was performed using Platinum SuperFi II DNA polymerase

(Thermo Fisher 12361010) to amplify the knock-in region using left and right homology arm designated primers according to the manufacturer's protocol (see Appendix Table S2). The amplified donor DNA was collected using Monarch PCR and DNA clean-up kit (NEB T1030L) according to the manufacturer's protocol.

## Microscope

Fixed-cell data were acquired on a custom wide-field inverted Nikon Ti-2 wide-field microscope or an upright Nikon Eclipse Ni microscope equipped with a 60×1.4NA oil immersion objective lens (Nikon), Spectra X LED light engine (Lumencor), and Orca 4.0 v2 scMOS camera (Hamamatsu). The x–y pixel size was 108.3 nm, and the z-step size was 300 nm.

Live-cell data were acquired with either TIRF or RING-TIRF set-up as described below.

## TIRF

Live-cell data was acquired on a custom inverted wide-field Nikon Eclipse Ti-E microscope equipped with three Andor iXon Ultra DU897 EMCCD cameras (512x512pixels), apochromatic TIRF ×100 oil immersion objective lens(1.49 NA, Nikon, MRD01991), linear encoded XY-stage with 150 micron travel range piezo z (Applied Scientific Instrumentation), and LU-n4 four laser unit with solid state 405, 488, 561, and 640 nm lasers (Nikon), a TRF89901-EM ET-405/488/561/640 nm laser quad band filter set for TIRF applications (Chroma), and Nikon H-TIRF system. The x–y pixel size was 160 nm.

## RING-TIRF

Live-cell data was acquired on a custom inverted wide-field Nikon Eclipse Ti-2E microscope equipped with ORCA-Fusion BT CMOS camera (6.5 μm × 6.5 μm), apochromatic TIRF ×60 oil immersion objective lens (1.49 NA, Nikon, MRD01691), LUN-F Multi laser unit with 405 nm, 488 nm, 561 nm, and 640 nm lasers, a TRF89901-EMV2 ET-405/488/561/640 nm laser quad band filter set for TIRF applications (Chroma), and iLas2 (GATACA) 360°/Ring/Azimuthal TIRF system at a TIRF penetration depth of 150 nm. The x–y pixel size was 108 nm. All microscopes were under the automated control of the Nikon Elements software.

## Live-cell PCP-PP7 imaging

In total, 150,000 U-2 OS cells stably expressing tdMCP-mScarlett-RFP-CAAX and PCP-HaloTag were seeded in 35-mm glass-bottom dishes (Cellvis, D35-20-1.5-N) ~48 h prior to imaging. The following day, media was exchanged for 2 mL of culture media. Each dish was transfected with a mixture of DNA composed of 100 ng of reporter plasmid and 500 ng of an empty plasmid to 2.4 μL X-tremeGENE HP DNA transfection reagent (Millipore Sigma 6366244001) in 55 μL 37 °C Opti-MEM Reduced Serum Medium (Gibco 31985062). The solution was gently mixed and incubated at room temperature for 15–20 min and then added dropwise to the sample. In all, 12–16 h following transfection, cells were dyed by removing 100 μl of media and adding 2 μL of 100 μM JFX-646 (a gift from Luke Lavis) for a final concentration of 100 nM of dye. Sixty minutes after adding dye, dishes were washed

three times with warm culture media and then returned to the incubator in culture media. This washing step was repeated every 10–15 min for 1 h. At the time of imaging, media was exchanged for FluoroBrite DMEM (Gibco A1896701) media without phenol red supplemented with 10% FBS and 1× GlutaMAX (Gibco 35050061). Imaging was performed using RING-TIRF imaging system 35%/65% power was used for 549/646 laser and exposed for 300 ms/500 ms, respectively. For steady-state live-cell movies, an image was captured every 2 s for 5 min duration.

For 1× PBS and 1× L4-PBS binding kinetics measurements, a single cell clone of the U-2 OS cells stably expressing tdMCP-mScarlett-RFP-CAAX and tdPCP-HaloTag was acquired. In total, 45,000 cells were seeded in 35-mm glass-bottom dishes ~48 h prior to imaging. In all, 300 ng of reporter plasmid and 300 ng of empty plasmids were incubated in 2.4 μL X-tremeGENE HP DNA transfection reagent in 55 μL of 37 °C Opti-MEM Reduced Serum Medium for 30 min. In all, 12–16 h after transfection, cells were incubated with 2 μL of 1 μM JFX-646 for an hour with subsequent washing steps as priorly stated. Imaging was performed using RING-TIRF imaging system with 561 nm (50 mW) and 650 nm (40 mW) lasers at 20% for 561 nm laser and 5%,10%, 20%, or 40% for 646 nm laser and exposed for 200 ms/100 ms, respectively.

## Fixed-cell ZNF598-HaloTag imaging

All solutions were prepared in nuclease-free water (Quality Biological 351-029-131CS). In all, 18 mm #1 coverslips (Fisher 12-545-100) were washed with 70% ethanol and rinsed 4× with PBS (Corning 21-031-CV). Overall, 40,000 U-2 OS cells were seeded on a coverslip with 1 mL of culture media. In all, 12–16 h following seeding, cells were dyed by removing 100 μl of media, adding 1 μL of 100 μM JFX-646 for a final concentration of 100 nM of dye. Thirty minutes after dyeing, cells were washed three times with warm culture media and then returned to incubator in culture media. This washing step was repeated two more times every 15 min. Cells were quickly rinsed 3× with PBS and fixed using 4% paraformaldehyde (Electron Microscopy Sciences 50-980-492) diluted in PBS. After 15 min of fixation, cells were rinsed 2× with PBS and allowed to incubate at room temperature for 10 min. Cells were permeablized using 0.1% of Triton x-100 (Millipore Sigma T8787-100mL) diluted in PBS for 10 min. Cells were rinsed 3× with PBS and allowed to incubate at room temperature for 10 min. Coverslips were then mounted on a pre-cleaned frosted glass cover slides (Fisher 12-552-3) using ProLong Diamond antifade mounting reagent with DAPI (Invitrogen P36962). Coverslips were allowed to cure for 18 h before imaging.

## Live-cell translation imaging

150,000 U-2 OS ZNF598-HaloTag Knock-in cells stably expressing tdMCP-mScarlett-RFP-CAAX and OSTIR-IRES-scFV-sfGFP were seeded in 35-mm glass-bottom dishes ~48 h prior to imaging. The following day, the media was exchanged for 2 mL of culture media. Each dish was transfected with 1 μg DNA to 4 μL X-tremeGENE HP DNA transfection reagent to 100 μL 37 °C Opti-MEM Reduced Serum Medium. The solution was gently mixed and incubated at room temperature for 15 min and then added dropwise to the sample. In all, 12–16 h following transfection, cells were dyed by removing 100 μl of media, adding 4 μL of 250 mM 3-indoleacteic

acid and adding 2 μL of 100 μM JFX-646 for a final concentration of 500 μM of IAA and 100 nM of dye. Sixty minutes after adding dye, dishes were washed three times with warm culture media and then returned to incubator in culture media supplemented with 500 μM IAA. This washing step was repeated every 10–15 min for 1.5–2 h. At the time of imaging, media was exchanged for FluoroBrite DMEM media without phenol red supplemented with 500 μM IAA, 10% FBS, 1× GlutaMAX.

When using RING-TIRF imaging system 20%/35%/65% power was used for 488/561/640 laser and exposed for 500 ms/300 ms/500 ms, respectively.

When using TIRF imaging system 4%/5% power was used for 488/561 laser and exposed for 500 ms for both lasers.

## Harringtonine runoff experiments

Cells were prepared as outlined in "Live cell translation imaging". Runoff experiments were performed as outlined in Livingston et al, 2023 with slight modifications. After finding a position, we removed 750 μL of media from the sample dish and mixed in a tube containing harringtonine. The mixture was mixed quickly, returned to imaging dish for a final harringtonine concentration of 9 mg/mL, and thoroughly mixed in imaging dish. The data were acquired at a 10 or 15-s frame rate for 30 min 1 min after adding harringtonine.

## Immunoblotting

U-2-OS cells were gently washed with PBS and lysed in ice-cold RIPA buffer containing 50 mM Tris (pH 7.5), 150 mM NaCl, 0.1% w/v SDS, 1% v/v NP40, 100 mM NaF, 17.5 mM β-glycerophosphate, 0.5% v/v sodium deoxycholate, 10% v/v glycerol, 2 mM $Na_3VO_4$, supplemented with fresh 1 mM PMSF, and Protease Inhibitor Cocktail (0.5% v/v, Millipore Sigma, P8340). After sonication on ice, the samples were centrifuged at $12,000 \times g$ for 10 min at 4 °C, and supernatant were collected and suspended in 1× SDS loading buffer. Samples were further incubated on a thermal mixer at 95 °C, 600 rpm for 10 min. Following SDS-PAGE and western blotting, membranes were incubated with primary antibodies at 4 °C overnight, including anti-β-tubulin (ProteinTech, 10094-1-AP) and anti-ZNF598 (Thermo Fisher, 703601). After thorough washes with TBST, the membranes were either incubated with a fluorescence secondary antibody at room temperature for 2 h (anti-β-tubulin) or with an HRP-linked antibody at 4 °C overnight (anti-ZNF598). Western blot images were either captured by an Odyssey scanner (LI-COR) for fluorescence imaging or by an GeneGnome XRQ scanner (SYNGENE) for chemiluminescence imaging. Protein expression levels were normalized with β-tubulin protein level, and the images were analyzed with ImageJ (version 2.3.0).

## Genome PCR amplification

DNA was extracted from U-2 OS cells using DNA extraction kit (Qiagen 69504) following the manufacturer's protocol. DNA was eluted in 30 μL of water. After extraction, PCR was performed using Platinum SuperFi II DNA polymerase to amplify the knock-in region using designated primers following the manufacturer's protocol (see Appendix Table S2). PCR was run in a 1% agarose gel (Invitrogen 16500100) prepared with TAE Buffer (Quality Biological 351-008-131) supplemented with Ethidium Bromide (Invitrogen 15585011). Gel electrophoresis was performed for 30 min at 500mAmp and 130 V. Correct knock-in band size was cut and extracted from gel using Monarch Gel clean-up (NEB T1020S) following manufactures recommendations. Amplified DNA was sent for Sanger sequencing.

## Harringtonine assay

Cells were prepared as outlined in "Live cell translation imaging". At the time of imaging, samples were placed on the microscope and allowed to stabilize at 37 °C. Positively transfected cells containing 15–50 membrane-tethered mRNAs and translation sites were identified for imaging as the samples came up to temperature equilibrium. Cells were imaged as outlined in "RING-TIRF imaging" every 10 s for 2 min. After steady-state-imaging, 750 μL of media was removed from the sample dish and harringtonine was added to a concentration of 9 mg/mL and mixed thoroughly in the imaging dish. Ten minutes after Harringtonine addition, cells were imaged again every 10 s for 2 min.

## smFISH probe labeling

To label probes for smFISH, a protocol was adapted from (Gaspar et al, 2017). Briefly, 48 20-mer oligonucleotides (see Appendix Table S3) targeting SunTag mRNA were pooled and conjugated to amino 11-ddUTP at the 3′ end using terminal deoxynucleotidyl transferase (TdT). Reaction was allowed to incubate overnight at 37 °C. Reaction was quenched with sodium acetate (final concentration 300 mM), and sample was loaded onto a Spin-X centrifuge column loaded with Bio Gel P-4 beads (Bio Rad, 1504124) for size-exclusion purification. Purified oligos were then labeled with Cy3-NHS ester (Lumiprobe, 41020), reaction was allowed to go overnight at room temperature. Following labeling, the probes were purified to remove unconjugated dye through Spin-X centrifuge column load with P-4 beads. Labeling efficiency and yield were determined by nanodrop.

## smFISH-immunofluorescence (smFISH-IF)

smFISH-IF was performed as described in Livingston et al, 2023. In total, 45,000 U2TF cells stably expressing OSTIR-IRES-scFV-sfGFP with either doxycycline inducible 24xSunTag-AID-24xMS2 or 24xSunTag-A60-AID-24xMS2 mRNA reporters were seeded on 18 mm #1 coverslips (Fisher 12-545-100). The next day, the media was changed with the addition of 500 μM IAA. On day 3, cells were induced for either one or 2 h with 1 μg/mL doxycycline. For UV-treated samples, cells underwent 500 J/m² and were allowed to recover for 15 min at 37 °C before fixation with 4% paraformaldehyde in molecular biology grade water (Quality Biology, 351-029-131) with 1× PBS (Corning, Cat. #: 46-013-CM) and 5 mM magnesium chloride (Millipore Sigma, M2670-500G) (PBSM) for 10 min at room temperature. Cells were washed thrice with DPBS before being permeabilized at room temperature for 10 min with 0.1% TrixonX 100 (Millipore Sigma T8787-100mL), 10 U of Superase Inhibitor (Fisher, AM2694), 5 mg/mL of BSA (VWR, 0332-25 G), 2 mM of Ribonucleoside Vanadyl Complex (RVC | NEB, S1402S), in PBSM. Cell were washed thrice before incubating

for 30 min at room temperature in prehybridization buffer containing 10% deionized formamide (Millipore Sigma, F9037-100ML), 2× SSC (Corning, 46-020-CM), 5 mg/mL BSA (VWR), 10U Superase Inhibitor, and 2 mM RVC. Hybridization of the smFISH and primary anti-GFP antibody (Aves Labs, GFP-1010, 1:1000 dilution) was performed for 3 h at 37 °C in hybrization solution containing 10% deionized formamide, 10% w/v dextran sulfate (Millipore Sigma, D8906-100G), 5 mg/mL of Ultrapure BSA (Invitrogen, AM2616), 2× SSC, 2 mM RVC, 1 mg/mL of FISH RNA competitor Salmon Sperm (Invitrogen, 15632011 V), and either SunTag and/or MS2 smFISH probes Cy3 and Atto590 labeled accordingly. Cells were then washed thrice for 5 min each with 10% deionized formamide and 2× SSC in molecular grade biology water warmed to 37 °C. Secondary antibody labeling was performed twice, incubating at 37 °C for 20 min each with Alexa-488-goat anti-chicken secondary antibody (Thermo Fisher, Cat. #: A-11039, 1:1000 dilution) in 10% formamide and 2× SSC in molecular grade biology water. Finally, cells were washed thrice for 10 min each with 1× PBS at 37 °C before being mounted under ProLong Diamond Antifade mount containing DAPI (Invitrogen, Cat. #: P36962) on frosted glass slides (Thermo Fisher, 12-552-3). The next day, coverslips were sealed with clear nail polish.

## UV damage assay

Cells were prepared as outlined in "Live cell translation imaging". At the time of imaging, samples were placed on the microscope and allowed to stabilize at 37 °C. Positively transfected cells containing 15–50 membrane-tethered mRNAs and translation sites were identified for imaging as the samples came up to temperature equilibrium. Cells were imaged as outlined in "RING-TIRF imaging" every 10 s for 2 min. After steady-state-imaging, the plate lid was removed, and samples underwent 500 J/m² UV-C exposure using UV StrataLinker 2400. The sample was set back on the microscope and allowed temperature to equilibrate to 37 °C. Positively transfected cells containing 10–50 membrane-tethered mRNAs and translation sites were identified for imaging. Cells were imaged as outlined in "RING-TIRF imaging" every 10 s for 2 min.

## Quantification and statistical analysis

### Analysis of live-cell imaging data

Datasets were analyzed using a combination of custom MATLAB code, Airlocalize (Lionnet et al, 2011), u-track (Jaqaman et al, 2008) and step detection algorithm Tdetector (Chen et al, 2014). Particle detection was performed using AirLocalize, while u-track was used for tracking. For steady state, imaging experiments mRNA tracks shorter than 25 frames were discarded and tracks shorter than four frames of ZNF598 were discarded. A temporal overlap of at least four frames was required in order to link mRNA and ZNF598/SunTag tracks. Only tracks lasting at least 2 min were included in the analysis. For harringtonine and UV damage assay imaging experiments, only tracks ≥2 min of imaging were included in the analysis. A temporal overlap of at least 10 s (or 2 frames) was required in order to link mRNA and ZNF598/SunTag tracks. For all movies, mRNAs for which tracking was disrupted due to crossing paths with another mRNA were discarded. Overlap of the tagged protein (either PCP-HaloTag or ZNF598-HaloTag) with mRNA was detected manually using custom MATLAB code

(TrackViewer). Protein "on" states were manually defined using custom MATLAB code (step detection) which incorporated the Tdetector code. Sample sizes (number of cells and mRNA molecules) are indicated in figure legends.

To calculate the integrated intensity of the tagged protein (either PCP-HaloTag or ZNF598-HaloTag) custom MATLAB code was used to fit 2D Gaussian to calculate the integrated spot intensity. The mean integrated intensity of the 1× PBS "bound" was used to normalize the integrated intensity of 1× PBS, 3× PBS, 12× PBS and ZNF598-HaloTag.

To calculate the fraction of protein colocalizing to mRNA per cell, the tagged protein (either PCP-HaloTag or ZNF598-HaloTag) were considered colocalizing to mRNA if there were at least five consecutive frames (~10 s overlap) overlapping to mRNA or two frames (~10 s overlap) for Harringtonine and UV assay. Fraction of colocalization analysis only included cells >10 translating mRNAs. Integrated intensity analysis only included protein–mRNA interactions that met the colocalization criteria to mRNA.

### Analysis of live-cell harringtonine runoff data

Data were analyzed using a combination of custom MATLAB code, Airlocalize (Lionnet et al, 2011) and u-track (Jaqaman et al, 2008). Particle detection was performed using AirLocalize, while u-track was used for tracking. Tracks shorter than five frames were discarded. A temporal overlap of at least five frames was required in order to link mRNA and SunTag tracks. mRNAs for which tracking was disrupted due to crossing paths with another mRNA were discarded. To calculate the clearance time on each mRNA, similar analysis was performed as in Goldman et al, 2021. Briefly, we first determined when the SunTag signal reached its maximum value, and then found the time after this point at which the signal fell below 10% of its maximum intensity. We performed the same analysis for the mRNA channel. If loss of signal for mRNA and SunTag were coincident within three or fewer frames, we did not include the molecule for analysis, due to concerns about signal disappearance for reasons other than clearance of ribosomes (e.g., mRNAs leaving the membrane). Because the mRNA signal was more difficult to track for the full 30 min than the SunTag signal, we included molecules in the analysis for which SunTag signal persisted after the loss of the mRNA signal. mRNAs were not included if the mean signal of the first four frames was less than 10% of the maximum intensity (considered not to be translating at the start of the experiment). Sample sizes (number of cells and mRNA molecules) are indicated in figure legends. 95% confidence bounds were estimated using Greenwood's formula.

### Analysis of smFISH-IF data

smFISH-IF data was analyzed similarly as described in Livingston, et al 2023. Briefly, cells translating the reporter were outlined with FISH-Quant (Mueller et al, 2013). To analyze outlined cell images, a custom MALTAB code (uLocalize) was used to identify translation sites intensity for single mRNAs. Briefly, the mRNA channel image is filtered with Laplacian of Gaussian filter to remove noise. The filtered image is used to identify candidate spots, which are quantified using Gaussian fitting to identify location coordinates and intensity. The location of the RNA spot is used to form a region of interest (ROI) in the protein channel for potential translation site detection. To detect and analyze the mature single proteins, the ROI containing potential translation sites was

excluded. The protein channel is analyzed similarly to the RNA channel: the positions and the integrated intensities of single proteins are identified and determined through Gaussian Fitting. To calculate the integrated intensity of translation sites, the corresponding ROI is fitted with 3D Gaussian. To reduce picking up background signal, we imposed a minimal threshold of amplitude and a maximal distance threshold to the corresponding mRNA. Finally, the integrated intensity of translation site is normalized to the single-protein intensity in the same cell to calculate the number of nascent peptides. The program can be found on the Wu Lab Github site (https://github.com/binwulab/uLocalize.git).

To test distributions of number of ribosomes per mRNA for statistical significance, distributions were first transformed by taking the square root of all values to reduce skewness. The two-tailed sample *t* test was then applied to the transformed distributions.

## Data availability

Custom code for analysis of live-cell imaging data is available at https://github.com/binwulab.

The source data of this paper are collected in the following database record: biostudies:S-SCDT-10_1038-S44318-025-00523-z.

## Peer review information

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

## Acknowledgements

This study is supported by the National Institutes of Health (R01GM138770, RF1NS113820) and National Science Foundation (MCB1817447) to BW, and HHMI for RG. ACD is supported by the National Science Foundation Graduate Research Fellowship (Grant# DGE-1746891), ACD and GT are supported by NIH training grant T32-GM135131, ZH is supported by NIH training grant T32 GM007445. We thank Dr. Luke Lavis at the Janelia Research Campus for the fluorescent dyes used in this study.

## Author contributions

**Ana C De La Cruz**: Conceptualization; Data curation; Software; Formal analysis; Validation; Investigation; Visualization; Methodology; Writing—original draft; Writing—review and editing. **Garrett Tisdale**: Software; Formal analysis; Validation; Investigation; Visualization; Methodology; Writing—review and editing. **Emily Nakayama**: Formal analysis; Investigation; Visualization. **Zhiyuan Huang**: Data curation; Formal analysis; Investigation. **Niladri K Sinha**: Resources; Methodology. **Rachel Green**: Resources; Supervision; Investigation; Writing—review and editing. **Bin Wu**: Conceptualization; Resources; Supervision; Funding acquisition; Investigation; Writing—original draft; Project administration; Writing—review and editing.

Source data underlying figure panels in this paper may have individual authorship assigned. Where available, figure panel/source data authorship is listed in the following database record: biostudies:S-SCDT-10_1038-S44318-025-00523-z.

## Disclosure and competing interests statement

Rachel Green is a member of the Advisory Editorial Board of The *EMBO Journal*. This has no bearing on the editorial consideration of this article for publication.

