## [Peer Review File · The EMBO Journal]

Single-protein/RNA imaging reveals ZNF598 as a limiting factor in resolving collided ribosomes

Ana De La Cruz, Garrett Tisdale, Emily Nakayama, Zhiyuan Huang, Niladri Sinha, Rachel Green, and Bin WU

Corresponding author(s): Bin WU (bwu20@jhmi.edu)

Review Timeline:

Submission Date:	18th Oct 24
Editorial Decision:	10th Dec 24
Revision Received:	23rd Apr 25
Editorial Decision:	28th May 25
Revision Received:	10th Jun 25
Accepted:	7th Jul 25

Editor: *Cornelius Schneider*

Transaction Report:

Dear Dr. WU,

Thank you for submitting your manuscript for consideration by the EMBO Journal. It has now been seen by three referees whose comments are shown below.

As you can see from the reports all three referees agree that the manuscript contributes several interesting new aspects to the known role of ZNF598 in resolving collided ribosomes. In addition, the referees find that the manuscript provides an important methodological and technical contribution and we agree with referee #1 that it would benefit from a stronger focus on this part of the manuscript. Given the referees' positive recommendations, I would like to invite you to submit a revised version of the manuscript, addressing the comments of all three reviewers. I should add that it is EMBO Journal policy to allow only a single round of revision, and acceptance of your manuscript will therefore depend on the completeness of your responses in this revised version.

Thank you for the opportunity to consider your work for publication. I look forward to your revision.

Yours sincerely,

Cornelius Schneider, PhD
Editor
The EMBO Journal
c.schneider@embojournal.org

We realize that it is difficult to revise to a specific deadline. In the interest of protecting the conceptual advance provided by the work, we recommend a revision within 3 months (10th Mar 2025). Please discuss the revision progress ahead of this time with the editor if you require more time to complete the revisions. Use the link below to submit your revision:

XXXXXXXXXXXXXXXXXXXX

Referee #1:

Review of De La Cruz et al. "Single Protein-RNA imaging reveals ZNF598 as a limiting factor in resolving collided ribosomes"

The manuscript of De La Cruz et al. describes elegant live cell single-molecule imaging of the key ribosomal quality control factor ZNF598. The Wu and Green labs have previously investigated the roles of RQC in response to ribosomal elongational pausing, showing that polyA tracks lead to stalling and subsequent ubiquitinylation of the 40S ribosomal subunit through recognition of the unique collided ribosome conformation by ZNF598. However, many unanswered questions remain, including whether multiple queued ribosomes in a polysome are targeted during a pause, and how long and short timescale pauses are discriminated. The combined forces of the Wu and Green labs attempt to tackle these questions using real time, in live cell single molecule imaging to observe temporal colocalization of ZNF598 with translating ribosomes, gather stoichiometry information and delineate how cells respond to mRNA damage. The results presented here are generally beautifully performed and controlled, and the conclusions are warranted. As such, this is an excellent potential contribution for EMBO Journal.

My one major critique of the paper is that it has two main goals, which sometimes lead to the whole being less than the parts. First, the authors want to present methodological advances that demonstrate their ability to detect single protein binding to a single mRNA-this of course is a challenge in the context and background of the cell. The Wu lab has pioneered these approaches by tethering mRNAs to cell membranes to allow long timescale observation of translational phenomena on the mRNA with single-molecule resolution. Here they demonstrate single protein sensitivity by designing model mRNAs with multiple MS2 coat protein binding sites, and detect the transient binding of multiple copies of the protein compared to a control without a binding site. Here the data are clear, but as this is the benchmarking of their ability to gain both stoichiometries and residence times, I would have liked to see deeper analysis-are the intensity changes really protein binding or are some of the fluctuations photophysical? Can the authors gain bound state lifetime or arrival kinetics? What occurs if you weaken the MS2-mRNA affinity? Maybe I am old fashioned but a bit deeper characterization of the in vivo signals vis a vis physical properties would build the foundation of this technology. If the authors can fit perhaps an additional paragraph on the methodology (without the constraints of journal article length) that would greatly strengthen the subsequent investigation of ZNF598.

To explore the function of ZNF598 during stalling, the authors build on the technology discussed above, but include now a polyA stalling sequence known to cause ribosomal collisions during elongation. The authors endogeneously tagged ZNF598 using CRISPR to allow tracking and demonstrate that the tagged version is functional in vivo. They then observe the stoichiometry of ZNF598 on translating mRNAs, showing that mRNAs with collision sequences have more than one molecule bound, suggesting that multiple collided ribosomes per mRNA are detected by the protein. This opens up a novel vista on the mechanism, as it was previously hypothesized that ZNF598 binds only the leading collided ribosome. Since ZNF598 concentrations are low compared to total ribosomal pool, the authors then show faster resolution of collided ribosomes upon its overexpression, also supporting that ubiquitinylation is rate limiting the the RQC process. Surprisingly they observe a large fraction of normal (untagged) mRNAs that are bound by ZNF598, suggesting a larger pool of damaged mRNAs that cause pausing than previously imagined. There are many such intriguing observations presented here, that will surely be the foundation of future mechanistic work by the groups.

In summary, this is outstanding in vivo biophysics, probing an important question in translation biology. The manuscript suffers from presenting methods and some initial observations on ZNF598. Deepening the methods section to buttress the conclusions of the ZNF598 portion will make this an even better manuscript.

Referee #2:

In this manuscript, De La Cruz et al. present a imaging approach to visualize the recruitment of individual protein molecules to single mRNAs. Using this approach, the authors study the recruitment of Halo-tagged ZNF598, a sensor of collided ribosomes, to MS2-labeled mRNAs containing poly(A) pause sequences or chemical modifications caused by UV radiation. To simultaneously visualize the translation status of individual mRNAs, the well-established SunTag translation imaging system is employed.

After introducing the experimental setup, the authors employ their technique to show that multiple ZNF598 molecules can be recruited to a single poly(A)₆₀ mRNA, providing evidence that ZNF598 can bind to any collided ribosome in a long queue. Quantifying the number of ZNF598 per mRNA and comparing that number to the number of collided ribosomes showed that the large majority of collided ribosomes is not bound to ZNF598, suggesting that ZNF598 recruitment is a rate-limiting step in recycling of collided ribosomes. Indeed, ZNF598 overexpression experiments showed that ZNF598 is rate-limiting for the clearance of ribosomes on mRNAs. Finally, UV radiation experiments are done which provide further evidence that ZNF598 is a limiting factor in recognizing and resolving ribosome collisions under conditions of high collision load in the cell. Overall, this is an impressive study which uncovers binding dynamics of individual protein molecules to single reporter mRNAs in living cells, an important advance in the field. The experimental approach developed by the authors is likely to find applications in other studies into RNA-protein interactions in future research. The biological findings are also solid, albeit not very surprising, and the study is rigorous from a technical perspective. Overall, I support publication if the concerns below are adequately addressed.

Comments:

1. Figure 4D shows that ribosome runoff rates are higher when cells are overexpressing ZNF598, and it is argued that this change in runoff dynamics is the result of increased ribosome recycling rates when ZNF598 is overexpressed. Did the authors also examine the number of ribosomes per poly(A) mRNA with and without ZNF598 overexpression? It is possible that ribosome load is different in cells overexpressing ZNF598 and that the altered (i.e. lower) ribosomal load after ZNF598 overexpression affects run off rates?
2. Figure 5 presents data which shows that a large fraction of Harringtonine resistant translation sites is bound by ZNF598 10 minutes after Harringtonine addition. However, 10 minutes after Harringtonine addition, the total number of cellular ribosome collisions (both persistent and transient) is likely reduced, potentially freeing up ZNF598 and allowing ZNF598 to more efficiently bind those transcripts that still contain ribosomes. To confirm their conclusions that harringtonine resistant mRNAs show higher ZNF598 binding, can the authors compare ZNF598 levels between harringtonine resistant and non-resistant mRNAs BEFORE Harringtonine addition to assess whether ZNF598 occupancy is already increased on transcript that will fail to runoff once Harringtonine is added?
3. The authors comment several times that it's unknown whether ZNF598 is only binding the leading ribosome or also trailing ribosomes in a queue. Is there anything in the literature that would suggest that ZNF598 binds only to the leading ribosome? I thought ZNF598 simply binds to disomes, so I'm not sure where the hypothesis comes from that it only binds to the leading ribosome?
4. The authors state that binding of multiple ZNF598 molecules per poly(A) reporter mRNA proves that ZNF598 also binds to trailing ribosomes, not just to the leading one. Since it's not known how ZNF598 binds to collided ribosomes (to my knowledge), is it possible that multiple molecules of ZNF598 bind to the leading collision instead?
5. The authors state: "Overall, we developed an innovative assay to track single protein-mRNA interactions and their translation status in real time"
While the ability to see single molecules binding to a mRNA is very impressive, I don't know if the assay used to achieve this is in itself "innovative". They basically combine mRNA membrane tethering, which they have reported previously with TIRFM, a widely established microscopy method.
6. For cells expressing reporter mRNAs with 1xPP7, about half of the mRNAs are positive for PP7, suggesting that under their experimental conditions, about 50% of PP7 binding sites are occupied. For the 3xPP7 and the 12xPP7 one would therefore expect on average 1.5 and 6 PP7 molecules bind per mRNA. However, the mean Halo fluorescence intensity of the 3x and 12xPP7 reporter mRNAs appear somewhat lower than expected. Do the authors have an explanation for this?
7. After DNA damage, there's less ZNF598 found on reporter mRNAs with a poly(A) sequence. Can this be due to fewer ribosomes/collisions per mRNA after DNA damage? Have the authors looked at ribosome queue length before and after DNA damage?
8. In Figure 3D, it appears that all tracked poly(A)₆₀ mRNAs recruit ZNF598 at some point during the 5-minute time window, especially when comparing to the No-insert control. However, in Figure 3E, only around 80% of mRNAs were found to co-localize ZNF598-Halo during that time. Can the authors comment on this apparent discrepancy?
9. The Figure legend of Figure 5 is 'Harringtonine resistant translation sites are targets of RQC'. Although it is very probable that ZNF598 binding to Harringtonine-resistant translation sites results in RQC, no experimental evidence for the activation of the

RQC pathway is provided in the manuscript. The claim of the figure legends title should therefore be renamed to instead emphasize that Harringtonine-resistant transcripts are recognized by ZNF598.

Referee #3:

The study from Cruz et al demonstrates how the interactions of single RBP proteins with single mRNAs can be detected and quantified in living cells. This is a significant achievement that opens up many possibilities. The careful calibration of signals and well-designed controls also highlights the power of the technique to quantify the impact of transient binding events on mRNA translation dynamics. The biology of ZNF598 is also of great interest, given its association with collided ribosomes. The authors show how to estimate the number of endogenously tagged ZNF598 proteins that associate with an mRNA and provide compelling evidence that ZNF598 is rate-limiting. This rate-limiting behavior is especially clear in the UV irradiation experiments, where ZNF598 is found, somewhat counterintuitively, to associate less with ribosome-slowed or stalled transcripts. The explanation that in this case ZNF598 is overtaxed is convincing. Overall, I think the study is well done and deserves publication.

Some suggestions before publication:

1. I think a key measurable is the number of ZNF598 proteins per collided ribosome. You mention there are likely many more queued ribosomes to bind to, but that may not be true if the number of collided or queued ribosomes is not that large on the mRNA (maybe the majority are slowed, but spaced out, for example). In principle, you could address this issue by estimating the number of ribosomes in a queue. For example, this might be done by looking carefully at the HT-induced run-off. It seems the run-off curves for individual poly(A) transcripts is very abrupt (e.g. Fig. 4C), which makes some sense if the ribosomes are essentially all queued up near the end of the transcript and then suddenly released when the jam clears. Based on the intensity before this rapid run-off, an estimate of the number of queued ribosomes could be made in comparison to the number of ZNF598 proteins. If you found, say, 1 in 10 of the queued ribosomes is bound by ZNF598, then that would provide quantifiable evidence ZNF598 is rate limiting. It would also allow you to estimate the absolute length of ribosomal queues just based on ZNF598 intensities (the number of bound ZNF598 proteins), which seems like it might be another powerful technology in and of itself (for example, looking at all the other ZNF598 binding events that are presumably to other endogenous mRNAs in the cell).

2. How generalizable is this approach? Should comment on this in the discussion. For example, while it seems very powerful for some RBPs, the tethering and immobilization requirement means it would be less useful in other cases, for example, for investigating RBPs that may associate with biomolecular condensates.

Minor suggestions:

3. Would be nice to show the whole cell in Fig. 1, to get a sense of where in the cell you are imaging and also give a researcher who is attempting to replicate these experiments a better understanding of what 'good' cells look like.

4. How much variability is there from cell to cell in endogenous ZNF598 levels? You could estimate that from your KI cells, just looking at the distribution of Halo-intensities. Then you'd expect run-offs to be slower in cells with less total ZNF598. I only ask because it seems there is a lot of cell-to-cell variability in these types of single-mRNA translation experiments. It would be good to know if total ZNF598 levels contribute to the variability seen in Harringtonine run-off experiments from other groups. Related, what is the estimated average elongation rates in run-off curves in Fig. 2G?

Response to reviewers

We want to thank the reviewers for their thoughtful and constructive comments and specific suggestions for improving this manuscript. We have performed additional experiments and analysis to address the reviewers' concerns. Below is a detailed point-by-point response to reviewers' comments.

Reviewer #1:

The manuscript of De La Cruz et al. describes elegant live cell single-molecule imaging of the key ribosomal quality control factor ZNF598. The Wu and Green labs have previously investigated the roles of RQC in response to ribosomal elongational pausing, showing that polyA tracks lead to stalling and subsequent ubiquitinylation of the 40S ribosomal subunit through recognition of the unique collided ribosome conformation by ZNF598. However, many unanswered questions remain, including whether multiple queued ribosomes in a polysome are targeted during a pause, and how long and short timescale pauses are discriminated. The combined forces of the Wu and Green labs attempt to tackle these questions using real time, in live cell single molecule imaging to observe temporal colocalization of ZNF598 with translating ribosomes, gather stoichiometry information and delineate how cells respond to mRNA damage. The results presented here are generally beautifully performed and controlled, and the conclusions are warranted. As such, this is an excellent potential contribution for EMBO Journal.

(1) My one major critique of the paper is that it has two main goals, which sometimes lead to the whole being less than the parts. First, the authors want to present methodological advances that demonstrate their ability to detect single protein binding to a single mRNA-this of course is a challenge in the context and background of the cell. The Wu lab has pioneered these approaches by tethering mRNAs to cell membranes to allow long timescale observation of translational phenomena on the mRNA with single-molecule resolution. Here they demonstrate single protein sensitivity by designing model mRNAs with multiple MS2 coat protein binding sites, and detect the transient binding of multiple copies of the protein compared to a control without a binding site. Here the data are clear, but as this is the benchmarking of their ability to gain both stoichiometries and residence times, I would have liked to see deeper analysis-are the intensity changes really protein binding or are some of the fluctuations photophysical? Can the authors gain bound state lifetime or arrival kinetics? What occurs if you weaken the MS2-mRNA affinity? Maybe I am old fashioned but a bit deeper characterization of the in vivo signals vis a vis physical properties would build the foundation of this technology. If the authors can fit perhaps an additional paragraph on the methodology (without the constraints of journal article length) that would greatly strengthen the subsequent investigation of ZNF598.

We appreciate the reviewer's approval of our technical achievement and biological findings. We agree that strengthening the methodology section would build a strong foundation for the manuscript. Deeper characterization of the simple model RNA binding protein PCP would "benchmark the stoichiometry and residence time". Therefore, we perform additional experiments and analysis to investigate the binding of tdPCP-HaloTag to the 1xPBS mRNA reporter. We found that tdPCP indeed showed intermittent "on" and "off" binding patterns (Fig. 1D, Movie S1). We focused on measuring the residence time of tdPCP on mRNA because it is an intrinsic parameter, while the "off" time depends on the unknown tdPCP concentration and labeling. The residence time followed an exponential function with an average of around 3.6s. To make sure that the observed on-and-off is not due to dye blinking or limited by

photobleaching, we varied the illumination laser powers. The distribution of residence

Fig. R1 (Fig. 1G) Binding time distribution of tdPCP-HaloTag on 1xPBS. The cumulative distribution function (CDF) of “residence time” of tdPCP-HaloTag on 1xPBS mRNA with varying laser power. Data was compiled from two independent experiments. (5%: 150, 191; 10%: 162, 118; 20%: 183, 155; 40%:145, 188 tracks).

time did not change when laser power was varied by 8-fold (Fig. R1).

To further validate our measurements, we mutated PBS to vary the binding affinity: a single G->U point mutation in the PP7 loop region decreases the affinity by one order of magnitude (PP7 wide-type (WT): AUAUGG, $K_d = 1\text{nM}$; L4 PP7 mutant: AUAUGU, $K_d = 13\text{nM}$) (Lim, 2002). We tracked tdPCP-HaloTag binding to WT PBS and L4 mutant reporter mRNAs. To capture the transient binding, we have to decrease the image acquisition time to 100 ms. As expected, the binding time of tdPCP on the L4 mutant is significantly reduced (Fig. R2, Movies S2). We conclude that the L4 mutation reduces the stability of the PCP-PBS complex, increasing the dissociation rate and the K_d . This experiment validates our method’s sensitivity to measure single-molecule

Fig. R2 (Appendix Figure S1)

Fig. R2 (Appendix Figure S1) Scheme of wide-type (WT) and L4 Mutant PBS sequences. B. Example tdPCP-HaloTag intensity traces over time for WT vs L4 PBS mRNAs. **C.** The cumulative distribution functions (CDF) of the residence times for tdPCP-HaloTag on WT and L4-mutant PBS. (Two biological replicates for each condition (WT: 67, 83 tracks; L4: 107, 93 tracks).

binding kinetics in live cells.

(2) To explore the function of ZNF598 during stalling, the authors build on the technology discussed above, but include now a polyA stalling sequence known to cause ribosomal collisions during elongation. The authors endogeneously tagged ZNF598 using CRISPR to allow tracking and demonstrate that the tagged version is functional in vivo. They then observe the stoichiometry of ZNF598 on translating mRNAs, showing that mRNAs with collision sequences have more than one molecule bound, suggesting that multiple collided ribosomes per mRNA are detected by the protein. This opens up a novel vista on the mechanism, as it was previously hypothesized that ZNF598 binds only the leading collided ribosome. Since ZNF598 concentrations are low compared to total ribosomal pool, the authors then show faster resolution of collided ribosomes upon its overexpression, also supporting that ubiquitinylation is rate limiting the RQC process. Surprisingly they observe a large fraction of normal (untagged) mRNAs that are bound by ZNF598, suggesting a larger pool of damaged mRNAs that cause pausing than previously imagined. There are many such intriguing observations presented here, that will surely be the foundation of future mechanistic work by the groups.

In summary, this is outstanding in vivo biophysics, probing an important question in translation biology. The manuscript suffers from presenting methods and some initial observations on ZNF598. Deepening the methods section to buttress the conclusions of the ZNF598 portion will make this an even better manuscript.

We appreciate the reviewer's positive evaluation of our work. We have expanded the method section to characterize the PCP-PBS interaction further (See the previous response), according to the suggestion.

Reviewer #2:

In this manuscript, De La Cruz et al. present a imaging approach to visualize the recruitment of individual protein molecules to single mRNAs. Using this approach, the authors study the recruitment of Halo-tagged ZNF598, a sensor of collided ribosomes, to MS2-labeled mRNAs containing poly(A) pause sequences or chemical modifications caused by UV radiation. To simultaneously visualize the translation status of individual mRNAs, the well-established SunTag translation imaging system is employed. After introducing the experimental setup, the authors employ their technique to show that multiple ZNF598 molecules can be recruited to a single poly(A)₆₀ mRNA, providing evidence that ZNF598 can bind to any collided ribosome in a long queue. Quantifying the number of ZNF598 per mRNA and comparing that number to the number of collided ribosomes showed that the large majority of collided ribosomes is not bound to ZNF598, suggesting that ZNF598 recruitment is a rate-limiting step in recycling of collided ribosomes. Indeed, ZNF598 overexpression experiments showed that ZNF598 is rate-limiting for the clearance of ribosomes on mRNAs. Finally, UV radiation experiments are done which provide further evidence that ZNF598 is a limiting factor in recognizing and resolving ribosome collisions under conditions of high collision load in the cell.

Overall, this is an impressive study which uncovers binding dynamics of individual protein molecules to single reporter mRNAs in living cells, an important advance in the field. The experimental approach developed by the authors is likely to find applications in other studies into RNA-protein interactions in future research. The biological findings are also solid, albeit not

very surprising, and the study is rigorous from a technical perspective. Overall, I support publication if the concerns below are adequately addressed.

Comments:

1. Figure 4D shows that ribosome runoff rates are higher when cells are overexpressing ZNF598, and it is argued that this change in runoff dynamics is the result of increased ribosome recycling rates when ZNF598 is overexpressed. Did the authors also examine the number of ribosomes per poly(A) mRNA with and without ZNF598 overexpression? It is possible that ribosome load is different in cells overexpressing ZNF598 and that the altered (i.e. lower) ribosomal load after ZNF598 overexpression affects run off rates?

We thank the reviewer for highlighting this critical aspect of data interpretation. If over-expressing ZNF598 accelerates the clearance of the ribosomal queue, one would expect the queue to shorten. As the reviewer points out, a shorter ribosomal queue may also explain faster clearance due to ZNF598 overexpression. To explore this, we performed smFISH-IF to measure the number of nascent peptides (NAPs) on reporter mRNAs in the ZNF598 overexpression (OE) cell line. Surprisingly, for both no-insert and polyA60 mRNAs, the number of NAPs did not change significantly when ZNF598 was overexpressed (**Fig. R3** or Appendix Fig. S4). The discrepancy may arise from the difference in how we measured the ribosome loading versus runoff. In the smFISH-IF experiments, the ribosome loading is measured at steady-state, where translation initiation, regular termination, and disassembly of collided ribosomes occur concurrently. In contrast, Harringtonine blocks translation initiation so that existing ribosomes can run off, which directly measures the ribosome elongation and clearance from mRNAs. Based on these results, we hypothesize that translation initiation remains faster than clearance, even under ZNF598 overexpression, keeping the RQC mRNA densely packed with ribosomes. As a result, the ribosome queue did not become shorter, even when the clearance speed was increased. Of course, without direct measurement of the translation initiation rate, this remains speculative. Nevertheless, this data indicates that the increased ribosome clearance speed by overexpressing ZNF598 is not simply due to a shorter ribosome queue.

We included this new data in Fig. S4A-B and discussed the interpretation in the Discussion section.

Fig. R3 (Appendix Figure S4)

Fig. R3 (Appendix Fig. S4): Numbers of Nascent Peptides (NAPs) on reporter mRNAs are similar in wild-type (WT) and ZNF598-overexpressing (OE) cell lines. A. Quantification of the number of NAPs per mRNA in wild-type (mean: no-insert=13.1; poly(A)60=39.7) vs ZNF598-overexpression (mean: no-insert=11.7; poly(A)60=38.7) cells in smFISH-IF experiment. p-values between WT vs OE were calculated by two-sample t test after correction of distribution skewness (see methods for details), No-insert mRNAs: 0.9695; poly(A)60 mRNAs: 0.2719. **B.** Fraction of mRNAs actively translating. Each dot represents one cell; black lines indicate the mean. mRNAs calculated to have >150 ribosomes are included in the rightmost bin. Data compiled from two independent experiments: 73–165 cells; 1391–4393 mRNAs per condition. p values between WT vs OE were calculated by a two-sample t-test, no-insert: 0.3328; poly(A)60: 0.5522.

(2). Figure 5 presents data which shows that a large fraction of Harringtonine resistant translation sites is bound by ZNF598 10 minutes after Harringtonine addition. However, 10 minutes after Harringtonine addition, the total number of cellular ribosome collisions (both persistent and transient) is likely reduced, potentially freeing up ZNF598 and allowing ZNF598 to more efficiently bind those transcripts that still contain ribosomes. To confirm their conclusions that harringtonine resistant mRNAs show higher ZNF598 binding, can the authors compare ZNF598 levels between harringtonine resistant and non-resistant mRNAs BEFORE Harringtonine addition to assess whether ZNF598 occupancy is already increased on transcript that will fail to runoff once Harringtonine is added?

We appreciate the reviewer's excellent suggestion. The suggested experiment requires real-time tracking of mRNA and ZNF598 in the ribosome runoff experiment. We have shown that in WT cells, approximately 10% of mRNAs experience abnormal elongation rates and are potential targets of ZNF598 (Fig. 2G). Because each run-off experiment only results in one or two Harringtonine-resistant mRNAs per cell, the throughput is very low. Therefore, we chose to image many cells to increase the number of events. We have performed a few experiments as suggested by the reviewer. Indeed, in those instances, the ZNF598-pre-bound mRNAs do have a higher chance of failing to runoff (Fig. R4), which is in agreement with the reviewer's expectation. The average run-off time for ZNF598-pre-bound mRNAs is 21.8 minutes compared to 7.2 minutes for ZNF598-unbound ones. We have now included the new data in Appendix Figure S5.

Fig. R4 (Appendix Figure S5) ZNF598-pre-bound mRNAs have slower runoff rates than ZNF598-unbound mRNAs. Composite of normalized ZNF598 tracks and their corresponding translation intensity trajectories during runoff experiment (4 cells; 67 tracks).

3. The authors comment several times that it's unknown whether ZNF598 is only binding the leading ribosome or also trailing ribosomes in a queue. Is there anything in the literature that would suggest that ZNF598 binds only to the leading ribosome? I thought ZNF598 simply binds to disomes, so I'm not sure where the hypothesis comes from that it only binds to the leading ribosome?

We thank the reviewer for helping us to clarify the important conceptual advancement of our paper. It is well established that ZNF598 binds to the interface between disome (Juzskiewicz *et al*, 2018). However, there is no direct evidence in the literature demonstrating that ZNF598 exclusively binds the leading ribosome. By contrast, multiple studies have shown that RQC machinery specifically targets the leading collided ribosome for rescue (Juzskiewicz *et al*, 2020). Our work demonstrates that persistent collision due to a strong stalling sequence leads to a long queue of ribosomes. ZNF598, although as a limiting factor, can recognize the trailing collided ribosomes and ubiquitinate them, so prepare them for rapid clearance. If only the leading collided ribosome could be targeted by ZNF598, the one-by-one rescue of the long ribosomal queue would be even slower, as shown by a recent study (Madern *et al*, 2025). We have clarified this point further in the Discuss section.

4. The authors state that binding of multiple ZNF598 molecules per poly(A) reporter mRNA proves that ZNF598 also binds to trailing ribosomes, not just to the leading one. Since it's not known how ZNF598 binds to collided ribosomes (to my knowledge), is it possible that multiple molecules of ZNF598 bind to the leading collision instead?

Previous structural studies demonstrated that ZNF598 recognizes the interface between collided ribosomes, concluding that one ZNF598 binds a di-some (Juzskiewicz

et al, 2018). We observed that more than one ZNF598 was recruited to a polyA60 mRNA. So we conclude that it must also target trailing collided ribosomes.

5. *The authors state: "Overall, we developed an innovative assay to track single protein-mRNA interactions and their translation status in real time"*

While the ability to see single molecules binding to a mRNA is very impressive, I don't know if the assay used to achieve this is in itself "innovative". They basically combine mRNA membrane tethering, which they have reported previously with TIRFM, a widely established microscopy method.

We appreciate the reviewer's comment that observing single protein molecules on specific translating mRNAs "is very impressive". We believe that building new assays by integrating existing technologies is still innovative. However, we respect the reviewer's comment and have removed the "innovative" claim from the sentence.

6. *For cells expressing reporter mRNAs with 1xPP7, about half of the mRNAs are positive for PP7, suggesting that under their experimental conditions, about 50% of PP7 binding sites are occupied. For the 3xPP7 and the 12xPP7 one would therefore expect on average 1.5 and 6 PP7 molecules bind per mRNA. However, the mean Halo fluorescence intensity of the 3x and 12xPP7 reporter mRNAs appear somewhat lower than expected. Do the authors have an explanation for this?*

We thank the reviewer for raising the important issue of interpreting the fluorescence intensity of HaloTag. At the request of Reviewer 1, we have acquired new data about the "on time" distribution of tdPCP binding on 1xPBS (Fig. 1G, Supplementary Movie 1-2, Appendix Figure S1). tdPCP showed an on-and-off binding pattern. In Figure 1E, an mRNA is considered to have a tdPCP occupied as long as there is at least one tdPCP colocalization event within a 5-minute imaging window. Since about 50% 1xPBS RNA track has a binding event, the percentage of time that a PBS is bound by tdPCP is less than 50%. That may explain why the fluorescence intensities of 3x and 12x PBS reporter mRNAs are less than expected. The precise stoichiometry measurement may be complicated due to the following factors. First, PBS may not fold into native PBS states 100%. Second, the labeling efficiency of tdPCP-HaloTag is not saturating. Third, the on-and-off binding of tdPCP to PBS may contribute to less-than-expected fluorescence intensity at any instant of time. Although the intensities of 3x and 12x PBS are more complex, the 1xPBS fluorescence intensity is reliable because there are no other complications. Because of these reasons, we refrain from stating the actual stoichiometry of ZNF598. Instead, we only claim that the number of ZNF598 on polyA60 mRNAs is larger than one.

7. *After DNA damage, there's less ZNF598 found on reporter mRNAs with a poly(A) sequence. Can this be due to fewer ribosomes/collisions per mRNA after DNA damage? Have the authors looked at ribosome queue length before and after DNA damage?*

We believe the reviewer meant to say RNA damage instead of "DNA" damage. This is an important control to demonstrate that ZNF598 is limiting. Following the reviewer's suggestion, we performed smFISH-IF experiments to measure the ribosome

queue length of poly(A)60 reporter mRNA before and after UV damage (**Fig. R5**). Our data indicates that after UV damage, the nascent peptides (NAPs) on polyA60 reporter

Fig. R5 (Appendix Fig. S6): nascent peptides (NAPs) on poly(A)60 mRNA increase under the UV condition. Left: the number of NAPs per mRNA under +/-UV conditions. Mean: -UV= 40.3; +UV= 48.1; Data compiled from two independent experiments. mRNAs containing more than 150 NAPs are included in the rightmost bin. -UV:117 cells, 5863 mRNAs; +UV=184 cells, 7631 mRNAs; p-values were calculated by two-sample t-test after correction of distribution skewness (see methods for details). Right: fraction of translating mRNAs. Each dot represents one cell; black lines indicate mean. p value calculated by two-sample t-test

mRNAs increase (mean # of NAPs: 48 +UV vs 40 -UV) while the fraction of translating mRNAs stays the same. This demonstrates that the reduced ZNF598 recruitment to polyA60 mRNAs after UV damage is not due to fewer ribosomes in the queue. The new data is reported in Appendix Figure S6.

8. In Figure 3D, it appears that all tracked poly(A)60 mRNAs recruit ZNF598 at some point during the 5-minute time window, especially when comparing to the No-insert control. However, in Figure 3E, only around 80% of mRNAs were found to co-localize ZNF598-Halo during that time. Can the authors comment on this apparent discrepancy?

We thank the reviewer for pointing out this discrepancy in our figures. In the code used to generate Figure 3D, the tracks are plotted from most to least bound by ZNF598. In the original figure, more cells were selected for the poly(A)60 condition, resulting in more mRNA tracks than the no-insert control. Due to the y-axis limit, tracks corresponding to mRNAs that did not recruit ZNF598 in the poly(A)60 condition were cut off, giving the impression that all tracked mRNAs were bound. This explains the difference between Figures 3D and 3E. We now have replotted Figure 3D with a similar number of total tracks for no-insert and poly(A)60, which showed around 80% of tracks containing bound ZNF598 for poly(A)60 mRNAs.

9. The Figure legend of Figure 5 is 'Harringtonine resistant translation sites are targets of RQC'. Although it is very probable that ZNF598 binding to Harringtonine-resistant translation sites results in RQC, no experimental evidence for the activation of the RQC pathway is provided in the manuscript. The claim of the figure legends title should therefore be renamed to instead emphasize that Harringtonine-resistant transcripts are recognized by ZNF598.

We thank the reviewer for improving the rigor of the manuscript. We have changed the

figure legend “Harringtonine resistant translation sites are recognized by ZNF598”

Reviewer #3:

The study from Cruz et al demonstrates how the interactions of single RBP proteins with single mRNAs can be detected and quantified in living cells. This is a significant achievement that opens up many possibilities. The careful calibration of signals and well-designed controls also highlights the power of the technique to quantify the impact of transient binding events on mRNA translation dynamics. The biology of ZNF598 is also of great interest, given its association with collided ribosomes. The authors show how to estimate the number of endogenously tagged ZNF598 proteins that associate with an mRNA and provide compelling evidence that ZNF598 is rate-limiting. This rate-limiting behavior is especially clear in the UV irradiation experiments, where ZNF598 is found, somewhat counterintuitively, to associate less with ribosome-slowed or stalled transcripts. The explanation that in this case ZNF598 is overtaxed is convincing. Overall, I think the study is well done and deserves publication.

We greatly appreciate the reviewer’s approval of our study.

Some suggestions before publication:

1. I think a key measurable is the number of ZNF598 proteins per collided ribosome. You mention there are likely many more queued ribosomes to bind to, but that may not be true if the number of collided or queued ribosomes is not that large on the mRNA (maybe the majority are slowed, but spaced out, for example). In principle, you could address this issue by estimating the number of ribosomes in a queue. For example, this might be done by looking carefully at the HT-induced run-off. It seems the run-off curves for individual poly(A) transcripts is very abrupt (e.g. Fig. 4C), which makes some sense if the ribosomes are essentially all queued up near the end of the transcript and then suddenly released when the jam clears. Based on the intensity before this rapid run-off, an estimate of the number of queued ribosomes could be made in comparison to the number of ZNF598 proteins. If you found, say, 1 in 10 of the queued ribosomes is bound by ZNF598, then that would provide quantifiable evidence ZNF598 is rate limiting. It would also allow you to estimate the absolute length of ribosomal queues just based on ZNF598 intensities (the number of bound ZNF598 proteins), which seems like it might be another powerful technology in and of itself (for example, looking at all the other ZNF598 binding events that are presumably to other endogenous mRNAs in the cell).

We thank the reviewer for this suggestion. It would be ideal if we can directly measure the length of the queue. However, measuring the number of ribosomes in the queue in live cells is quite challenging because the translation initiation is stochastic and the positions of ribosomes are random. There have been previous attempts to deconvolve the intensity trace by assuming a stereotypical intensity profile of a single ribosome (Boersma *et al*, 2019). Unfortunately, it is not applicable to our case. We used a very low laser excitation in live cell experiment to capture the wide dynamic range of translation sites, so the single peptide intensity is low and very noisy. More importantly, because there are ribosomal queues of various lengths, the translation intensity profile of a single ribosome as a function of time is not well-defined. Therefore, the deconvolution of poly(A)₆₀ translation intensity in live cells is not feasible at this

moment. Further development of imaging and analysis tools is required to fulfill the reviewer's vision.

Therefore, we chose to measure the number of nascent peptides (NAPs) with smFISH-IF in fixed cells, where we can capture 3D stacks and fit the intensity profiles of single peptides and translation sites. We estimated the average number of NAPs on single mRNAs to be 13 on the no-insert and 40 on the poly(A)₆₀ reporters. If we assume that ribosomes are evenly distributed on the no-insert mRNA, there would be at least 27 NAPs in the queue. Given that ZNF598 binds to a disome interface (Juszkiewicz *et al*, 2018), we conclude that ZNF598 did not saturate potential binding sites.

2. How generalizable is this approach? Should comment on this in the discussion. For example, while it seems very powerful for some RBPs, the tethering and immobilization requirement means it would be less useful in other cases, for example, for investigating RBPs that may associate with biomolecular condensates.

We appreciate the reviewer's suggestion and have added a discussion about the limitations of this approach. Because we use tethering and immobilization, we cannot study protein-RNA interaction in the nucleus or biomolecular condensates.

New Discussion Text: Despite these limitations, our technique is broadly applicable to studying protein-RNA interactions that are not significantly influenced by membrane tethering. We anticipate it will be effective for investigating many protein-mRNA interactions occurring in freely diffusing mRNAs within the cytoplasm. However, interactions that depend on phase separation, occur within membraneless organelles or involve a highly dynamic exchange between compartments may require alternative imaging strategies.

Minor suggestions:

3. Would be nice to show the whole cell in Fig. 1, to get a sense of where in the cell you are imaging and also give a researcher who is attempting to replicate these experiments a better understanding of what 'good' cells look like.

We appreciate the reviewer's suggestion and have reformatted Figure 1 to show a whole cell.

4. How much variability is there from cell to cell in endogenous ZNF598 levels? You could estimate that from your KI cells, just looking at the distribution of Halo-intensities. Then you'd expect run-offs to be slower in cells with less total ZNF598. I only ask because it seems there is a lot of cell-to-cell variability in these types of single-mRNA translation experiments. It would be good to know if total ZNF598 levels contribute to the variability seen in Harringtonine run-off experiments from other groups. Related, what is the estimated average elongation rates in run-off curves in Fig. 2G?

The knock-in (KI) cell line used in this study is a monoclonal line. Therefore, we expect ZNF598 endogenous expression levels to be relatively uniform across cells. However, there still could be variability due to stochastic gene expression. To assess

variability, we analyzed the cytoplasmic ZNF598-Halo intensities from 60 individual cells. While there is some variation, the range of intensities remains relatively narrow, suggesting that endogenous ZNF598 levels do not fluctuate significantly between cells. We acknowledge that ZNF598 concentration can influence ribosome runoff dynamics, as demonstrated in our overexpression experiments (Fig.4D in the original manuscript). However, the observed cell-to-cell variability in our single-mRNA experiment is largely dominated by the small number of events in each cell. Due to technical limitations, we are restricted to image cells with a small number of mRNAs (typically less than 50), because too many mRNAs lead to too many mature proteins and the depletion of scFv-GFP). Because of the small number, the measured fraction of translating mRNAs, the

Fig. R6: ZNF598 fluorescence intensity in Knock-in cells Quantifications of ZNF598-HaloTag fluorescent intensity in 60 cells. Each dot represents one cell; black lines indicate mean and SEM.

mean number of ribosomes on mRNAs, and the survival curve of the ribosome runoff experiment will fluctuate significantly between cells. In the ZNF598 KI cells, the observed average data, such as runoff and ribosome occupancy, align well with previously published data (Goldman *et al*, 2021; Livingston *et al*, 2023).

Regarding the estimated average elongation rates from the run-off curves in Figure 2G, we have calculated them as follows.

Condition	Median runoff time	ORF length (codons)	Elongation rates (codons/s)
WT no-insert	5.4min	1,25	3.9
KI no-insert	4.8min	1,25	4.4
WT poly(A)60	14.1min	1,28	1.5
KI poly(A)60	13.1min	1,28	1.6

However, for the poly(A)60 reporter, the majority of the runoff time is probably spent on clearing ribosomes in the queue. So the above elongation rate is just an effective elongation rate.

References:

- Boersma S, Khuperkar D, Verhagen BMP, Sonneveld S, Grimm JB, Lavis LD & Tanenbaum ME (2019) Multi-Color Single-Molecule Imaging Uncovers Extensive Heterogeneity in mRNA Decoding. *Cell* 178: 458-472.e19
- Goldman DH, Livingston NM, Movsik J, Wu B & Green R (2021) Live-cell imaging reveals kinetic determinants of quality control triggered by ribosome stalling. *Molecular Cell* 81: 1830-1840.e8
- Juszkiewicz S, Chandrasekaran V, Lin Z, Kraatz S, Ramakrishnan V & Hegde RS (2018) ZNF598 Is a Quality Control Sensor of Collided Ribosomes. *Molecular Cell* 72: 469-481.e7
- Juszkiewicz S, Speldewinde SH, Wan L, Svejstrup JQ & Hegde RS (2020) The ASC-1 Complex Disassembles Collided Ribosomes. *Molecular Cell* 79: 603-614.e8
- Lim F (2002) RNA recognition site of PP7 coat protein. *Nucleic Acids Research* 30: 4138–4144
- Livingston NM, Kwon J, Valera O, Saba JA, Sinha NK, Reddy P, Nelson B, Wolfe C, Ha T, Green R, *et al* (2023) Bursting translation on single mRNAs in live cells. *Molecular Cell* 83: 2276-2289.e11
- Madern MF, Yang S, Witteveen O, Segeren HA, Bauer M & Tanenbaum ME (2025) Long-term imaging of individual ribosomes reveals ribosome cooperativity in mRNA translation. *Cell* 188: 1896-1911.e24

Dear Dr. WU,

Thank you for submitting a revised version of your manuscript. Your study has now been seen by all original referees, who find that their previous concerns have been addressed and now recommend publication of the manuscript with the exception of referee #2 who asks for a more detailed discussion on the stoichiometry of ZNF598 binding to collided disomes. In addition to this textual edit which I think is reasonable there remain only a few mainly editorial points that have to be addressed before I can extend formal acceptance of the manuscript:

- Please remove the figures and track changes from the manuscript and place the legends below the References.
- As we are switching from a free-text author contribution statement towards a more formal statement based on Contributor Role Taxonomy (CRediT) terms, please remove the present Author Contribution section and instead specify each author's contribution(s) directly in the Author Information page of our submission system during upload of the final manuscript. See <https://casrai.org/credit/> for more information.
- In the "Disclosure and competing interests statement", please add the following disclaimer: "Rachel Green is a member of the Advisory Editorial Board of The EMBO Journal. This has no bearing on the editorial consideration of this article for publication."
- As we are switching from a free-text author contribution statement towards a more formal statement based on Contributor Role Taxonomy (CRediT) terms, please remove the present Author Contribution section and instead specify each author's contribution(s) directly in the Author Information page of our submission system during upload of the final manuscript. See <https://casrai.org/credit/> for more information.
- Please list all callouts sequentially; A callout seems to be missing for Fig. 6E; callouts for Table S1-S3 should be corrected to Appendix Table S1-S3
- Please rename the movie files to Movie EV1-EV13 with the corresponding callouts
- Section order should be corrected: Title page - Abstract & Keywords - Introduction - Results - Discussion - Methods - Data Availability - Acknowledgements - Disclosure and Competing Interests Statement - References - Figure Legends - Table(s) - Expanded View Figure Legends.
- APPENDIX 1 FILE WITH ToC: Appendix file needs to be in PDF format; title page should contain "Appendix for + ms title"
- Please upload the R&T TABLE as a separate file using the template from our GTA
- Figure Legends (main + EV):
 1. Please note that the exact p values are not provided in the legend of figure 1E
 2. Please note that the dotted borders are not defined in the legend of figures 1C, 3B, 6B.

With best regards,
Cornelius Schneider

Cornelius Schneider, PhD
Editor | The EMBO Journal
c.schneider@embojournal.org

Thank you for submitting a revised version of your manuscript. Your study has now been seen by all original referees, who find that their previous concerns have been addressed and now recommend publication of the manuscript with the exception of referee #2 who asks for a more detailed discussion on the stoichiometry of ZNF598 binding to collided disomes. In addition to this textual edit which I think is reasonable there remain only a few mainly editorial points that have to be addressed before I can extend formal acceptance of the manuscript:

- Please remove the figures and track changes from the manuscript and place the legends below the References.
- As we are switching from a free-text author contribution statement towards a more formal statement based on Contributor Role Taxonomy (CRediT) terms, please remove the present Author Contribution section and instead specify each author's contribution(s) directly in the Author Information page of our submission system during upload of the final manuscript. See <https://casrai.org/credit/> for more information.
- In the "Disclosure and competing interests statement", please add the following disclaimer: "Rachel Green is a member of the Advisory Editorial Board of The EMBO Journal. This has no bearing on the editorial consideration of this article for publication."
- As we are switching from a free-text author contribution statement towards a more formal statement based on Contributor Role Taxonomy (CRediT) terms, please remove the present Author Contribution section and instead specify each author's contribution(s) directly in the Author Information page of our submission system during upload of the final manuscript. See

https://casrai.org/credit/ for more information.

- Please list all callouts sequentially; A callout seems to be missing for Fig. 6E; callouts for Table S1-S3 should be corrected to Appendix Table S1-S3

- Please rename the movie files to Movie EV1-EV13 with the corresponding callouts

- Section order should be corrected: Title page - Abstract & Keywords - Introduction - Results - Discussion - Methods - Data Availability - Acknowledgements - Disclosure and Competing Interests Statement - References - Figure Legends - Table(s) - Expanded View Figure Legends.

- APPENDIX 1 FILE WITH ToC: Appendix file needs to be in PDF format; title page should contain "Appendix for + ms title"

{Please upload the R&T TABLE as a separate file using the template from our GTA
- Figure Legends (main + EV):

1. Please note that the exact p values are not provided in the legend of figure 1E

2. Please note that the dotted borders are not defined in the legend of figures 1C, 3B, 6B.

With best regards,
Cornelius Schneider

Cornelius Schneider, PhD
Editor | The EMBO Journal
c.schneider@embojournal.org

Use the link below to submit your revision:

xx

Referee #1:

The authors have rigorously addressed my methodological concerns in the revised manuscript. It is now suitable for publication in EMBO J.

Referee #2:

Most of my concerns have been well addressed. There is however one point that requires further attention.

point 4 (reviewer). The authors state that binding of multiple ZNF598 molecules per poly(A) reporter mRNA proves that ZNF598 also binds to trailing ribosomes, not just to the leading one. Since it's not known how ZNF598 binds to collided ribosomes (to my knowledge), is it possible that multiple molecules of ZNF598 bind to the leading collision instead?

Response authors:

Previous structural studies demonstrated that ZNF598 recognizes the interface between collided ribosomes, concluding that one ZNF598 binds a di-some (Juszkiewicz et al, 2018). We observed that more than one ZNF598 was recruited to a polyA60 mRNA. So we conclude that it must also target trailing collided ribosomes.

My response:

I don't understand the argument of the authors here how the referenced paper shows that only a single protein ZNF598 binds to each disome. To my knowledge, the referenced paper does not show ZNF598 structure in the collided disome structure, nor is ZNF598 biochemically characterized as a monomer in this study. Can the authors be sure that ZNF598 binds as a single protein copy to each disome? Is it not possible that ZNF598 either binds to the disome as a protein dimer, or even that ZNF598 has multiple distinct binding sites on a collided disome? This seems important to know and/or discuss, as the main novelty claim of the paper is that ZNF598 binds to multiple disome pairs in a ribosome queue, and this conclusion is dependent on ZNF598 being a monomer and having only a single binding site on the disome.

Referee #3:

The author's revision has addressed my major issues.

Response to reviewers

We thank all reviewers for endorsing our revised manuscript. We appreciate the insightful comments of Reviewer #2 regarding ZNF598 stoichiometry. We have now addressed this point in detail and incorporated the corresponding discussion into the revised Discussion section.

Referee #2:

Most of my concerns have been well addressed. There is however one point that requires further attention. point 4 (reviewer). The authors state that binding of multiple ZNF598 molecules per poly(A) reporter mRNA proves that ZNF598 also binds to trailing ribosomes, not just to the leading one. Since it's not known how ZNF598 binds to collided ribosomes (to my knowledge), is it possible that multiple molecules of ZNF598 bind to the leading collision instead?

Response authors:

Previous structural studies demonstrated that ZNF598 recognizes the interface between collided ribosomes, concluding that one ZNF598 binds a di-some (Juszkiewicz et al, 2018). We observed that more than one ZNF598 was recruited to a polyA60 mRNA. So we conclude that it must also target trailing collided ribosomes.

My response:

I don't understand the argument of the authors here how the referenced paper shows that only a single protein ZNF598 binds to each disome. To my knowledge, the referenced paper does not show ZNF598 structure in the collided disome structure, nor is ZNF598 biochemically characterized as a monomer in this study. Can the authors be sure that ZNF598 binds as a single protein copy to each disome? Is it not possible that ZNF598 either binds to the disome as a protein dimer, or even that ZNF598 has multiple distinct binding sites on a collided disome? This seems important to know and/or discuss, as the main novelty claim of the paper is that ZNF598 binds to multiple disome pairs in a ribosome queue, and this conclusion is dependent on ZNF598 being a monomer and having only a single binding site on the disome.

We appreciate the reviewer's request for clarification. As the reviewer correctly points out, the cryo-EM study by Juszkiewicz et al. (2018) resolved only the collided disome. The known primary target of ZNF598, eS10, is localized at the 40S subunit interface between collided ribosomes. But the structure itself does not include ZNF598. Our earlier wording ("Previous structural studies demonstrated that ZNF598 recognizes the interface ...") therefore overstated the structural evidence, and we apologize for that inaccuracy.

Nonetheless, several experimental findings support that ZNF598 may act on the trailing collided ribosome pairs, rather than exclusively on the leading one. (1) Biochemical ubiquitylation assay (Juszkiewicz et al., 2018, Fig. 2D). When recombinant ZNF598 was incubated with ribosome queues of defined length, "the proportion of eS10 that was ubiquitinated increased for complexes containing more ribosomes. For example, penta-ribosomes were more effectively ubiquitinated on a per-ribosome basis than di-ribosomes". The authors concluded that "trailing

ribosome in the queue can be ubiquitinated by ZNF598". (2) PAR-CLIP analysis (Garzia *et al*, 2017). ZNF598 cross-linked to multiple sites upstream of the stall sequence on RQC substrates, implying that the E3-ligase can associate with different ribosomes along a ribosomal queue.

A high-resolution structure of ZNF598 bound to a disome remains to be determined, and no data currently support the stable oligomerization of ZNF598. Taken together, the biochemical, cross-linking, and our imaging data support a model in which ZNF598 can engage trailing collided ribosomes. We have added these clarifications and caveats to the Discussion section.

References:

- Garzia A, Jafarnejad SM, Meyer C, Chapat C, Gogakos T, Morozov P, Amiri M, Shapiro M, Molina H, Tuschl T, *et al* (2017) The E3 ubiquitin ligase and RNA-binding protein ZNF598 orchestrates ribosome quality control of premature polyadenylated mRNAs. *Nat Commun* 8: 16056
- Juszkiewicz S, Chandrasekaran V, Lin Z, Kraatz S, Ramakrishnan V & Hegde RS (2018) ZNF598 Is a Quality Control Sensor of Collided Ribosomes. *Molecular Cell* 72: 469-481.e7

Dear Dr. WU,

I am pleased to inform you that your manuscript has been accepted for publication in the EMBO Journal.

Yours sincerely,

Cornelius Schneider, PhD
Editor
The EMBO Journal
c.schneider@embojournal.org
